# Forest loss and uncertain economic gains from industrial and garimpo mining in Brazilian municipalities

Sebastian Luckeneder [1] ✉, Victor Maus [1,2], Juliana Siqueira-Gay [3], Tamás Krisztin [4] & Michael Kuhn [5,6]

Environmental and social risks in mining regions often juxtapose promises of local economic growth. Brazil, a major global mineral supplier and conservation leader, has pursued resource-led development despite mining's threat to its forests. Yet, the efficacy of this development strategy is uncertain. In this study, we examine mining's contribution to deforestation and regional economic growth in Brazil. For garimpo mining concessions – originally small-scale and less rigorously regulated forms of informal mining – we identify substantial associations with elevated deforestation rates, highlighting the environmental risks of insufficient oversight. The economic benefits of mining are limited. Particularly for industrial mining, they are tied to fluctuations in global mineral prices. These findings challenge the perception that mining inherently drives sustained regional economic development. As global demand for minerals rises – particularly to support the energy transition – strategic mining investments must be revised to prioritise sustained local progress, nature conservation and community well-being.

Global mining activities cover more than 100,000 km² of land[1,2], leading to substantial environmental and social impacts both directly at the mine sites and in their surrounding areas[3–5]. Yet, minerals and fossil fuels are indispensable for various aspects of human society, such as housing, energy, transport, and communication infrastructure. This creates a tension between domestic economic opportunities and the associated social and environmental risks[6–8]. Consequently, there are fundamental uncertainties regarding the compatibility of mining with sustainable development[9,10].

Brazil is not only among the world's leading suppliers of iron, gold, copper, and bauxite but also an important hub for tropical forest conservation. Among the various adverse consequences associated with mining[11–16], mining-induced deforestation is especially pronounced in Brazil[17]. Forest loss is caused directly at the mine sites, such

as at the actual extent of open-pits and forest clearing for on-site mining facilities, waste rock dumps and tailings ponds, but also indirectly due to transport, storage, processing and energy infrastructure build-up outside designated mining areas, as well as population pull effects and related urban and agricultural expansion[3]. Alarmingly, the country's mining territory expanded from 187 thousand ha in 2005 to 351 thousand ha in 2020[18] (Fig. 1), posing threats to areas vital for forest conservation and biodiversity[17,19,20]. This is of particular concern due to the resultant loss of natural habitats and destruction of carbon sinks, which could undermine efforts to meet global climate targets[21].

While there is broad consensus about the environmental and social risks of mining in Brazil, opinions diverge regarding the potential economic advantages that mining could bring[22,23]. Historically,

¹Institute for Ecological Economics, Vienna University of Economics and Business (WU), Vienna, Austria. ²Novel Data Ecosystems for Sustainability Research Group, Advancing Systems Analysis, International Institute for Applied Systems Analysis (IIASA), Laxenburg, Austria. ³Escola Politécnica, University of São Paulo, São Paulo, Brazil. ⁴Integrated Biosphere Futures Research Group, Biodiversity and Natural Resources, International Institute for Applied Systems Analysis (IIASA), Laxenburg, Austria. ⁵Economic Frontiers Program, International Institute for Applied Systems Analysis (IIASA), Laxenburg, Austria. ⁶Wittgenstein Centre, Vienna, Austria. ✉e-mail: sebastian.luckeneder@wu.ac.at

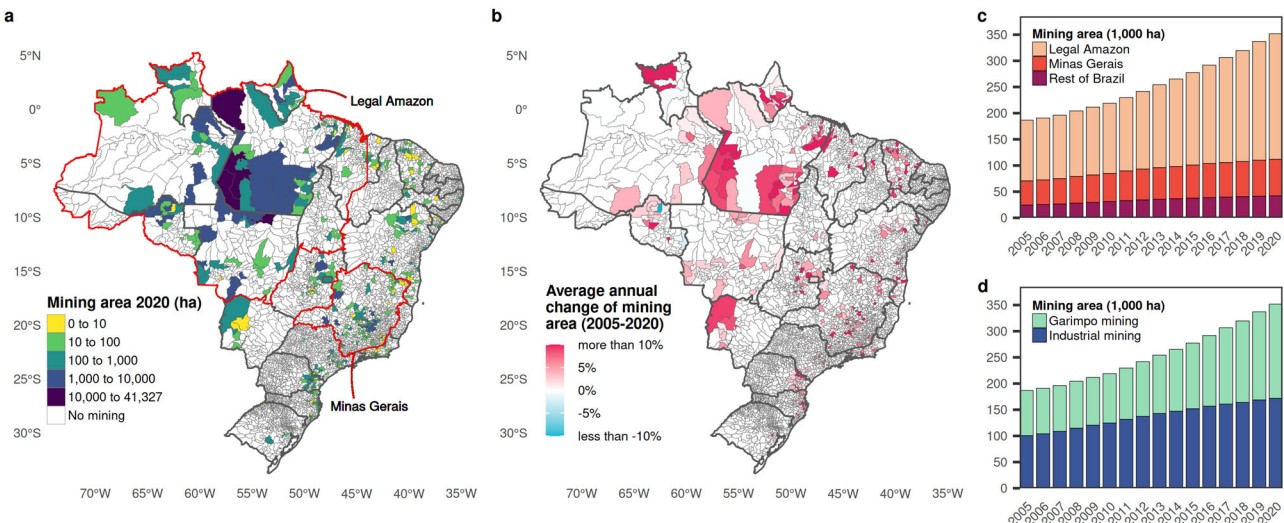

**Fig. 1 | Mining area by municipality and aggregated totals in Brazil, 2005-2020.** Map of total area classified as mining[18] per municipality in 2020 (in ha), with the Legal Amazon and Minas Gerais outlined in red (**a**); average annual change in mining area within municipalities between 2005 and 2020 (in%) (**b**); mining area by region (2005–2020, in 1000 ha) (**c**); and mining area by mining regulation type (2005–2020, in 1000 ha) (**d**). Basemaps are from the Brazilian Institute of Geography and Statistics, made available under a CC-BY 4.0 license[81], accessed via the `geobr` R package[75].

Brazil's extractive sector and institutions have portrayed both industrial and informal mining as pivotal to the economic development of its resource-rich peripheral territories[22], often in forested remote regions. Some studies support this view, showing that mining can spur regional economic development through job creation, local procurement, and associated local spillovers[24–26]. However, this perspective is countered by research on the resource curse thesis, which highlights the potential negative economic consequences of resource wealth. Studies suggest that in contexts of weak governance and institutional quality, an abundance of natural resources may hinder the development of key sectors such as manufacturing, education, and health – sectors which are vital for sustained growth[27–31]. Notably, for Brazil, empirical evidence remains inconclusive as to whether mining catalyses sustained local economic development.

This study investigates the contribution of mining to deforestation and regional economic growth in Brazil at the municipality level. The growing recognition of mining's localised environmental and socioeconomic consequences is underscored by recent research, such as Lo et al. (2024), which analysed the effects of nickel mining in Indonesia on forest cover and well-being[32]. Moreover, the present study differentiates between industrial and artisanal and small-scale mining, known as garimpos (Fig. 1d), both provided for in Brazilian legislation (Supplementary Notes A). By doing so, it offers a focused analysis that contrasts with broader assessments of macroeconomic mining impacts on national indicators[28,33]. A regional focus is warranted to understand the extent to which economic activities, such as mining, benefit the local economies that constitute the concrete living environment of the population. Moreover, it is warranted as a means to understand the spatial distribution of the gains and losses from such activities. We employed panel-structure Bayesian spatial econometric models, regressing 5-year average annual Gross Domestic Product (GDP) growth rates on a set of determinants of economic growth, augmented with land cover information and mine locations. The same setup was replicated to assess the effects of mining on forest loss. Besides its advantage of explicitly accounting for and evaluating the spatial spillovers that are central to this work, our spatial econometric approach offered a robust statistical framework that considered various national and local drivers of regional GDP and forest loss. All models relied on a consistent set of georeferenced data, sourced from remotely sensed land cover data products[18], Brazil's socioeconomic

statistics[34–36] and biophysical records[37,38], spanning 5262 municipalities from 2005 to 2020. To address the risk of bias from unobserved heterogeneity influencing where mining occurs in the first place, we applied a statistical matching algorithm, tentatively pruning the raw data to approximate a quasi-random distribution of treatment (mining) and control groups across the sample. Our findings indicate substantial associations between garimpo mining and increased deforestation, while for industrial mining, we do not observe a strong connection to forest cover dynamics. Regarding regional economic outcomes, the potential for positive effects appear more pronounced for industrial mining. However, neither garimpo nor industrial mining prompts a reliable or lasting increase in local GDP, with industrial mining even linked to negative effects in some years.

## Results
### GDP trends in mining regions
The Brazilian economy is widely influenced by mineral extraction. Throughout the study period, the economic growth of all Brazilian municipalities showed a strong correlation with global metals and minerals prices. However, our focus diverged from this overarching relationship between mining and the country's economic performance as we explored regional disparities observed between mining and non-mining municipalities. Descriptive statistics suggest nuanced dynamics between these two categories, as mining municipalities tend to have outperformed non-mining municipalities during periods of ascending commodity prices and vice versa (Fig. 2). Differences were notable in 2004 and 2010, when increases in metals and minerals prices coincided with 4.0 percentage points (pp) and 3.1 pp higher GDP growth rates, respectively, in mining as compared to non-mining municipalities, and in 2015 and 2016, when mining municipalities fell back by 2.6 pp and 2.8 pp, respectively, suggesting greater vulnerability to the recessionary pressures within these areas. Nevertheless, these observations necessitated further scrutiny within a controlled modelling framework, isolating the effects of mining at the municipality level by accounting for national macroeconomic trends and potential overarching and regional confounding factors.

Figure 3 presents estimates of the relative effect of mining, expressed as the differences in GDP growth (in pp) between mining and non-mining regions. These estimates are based on matched samples of mining (treated) and non-mining (control) municipalities,

derived from an econometric framework that accounts for several observed drivers of economic growth. The analysis further considers broader, unobserved economic factors – such as national policy schemes and world market commodity prices – that may have affected municipalities regardless of their mining exposure (see Methods for details). We used the presence of industrial and garimpo mining within a municipality as the mining indicator to facilitate the interpretation of results. Each panel shows mean effect estimates surrounded by 95% certainty intervals (CI) for the respective years or multi-year periods (pooled effects for pre-2010 and post-2010 time frames) in two colours. The darker colour represents direct effect within mining municipalities and the lighter colour indicates the magnitude of spillover effects of mining across municipality borders. All effects indicate estimated average differences as compared to non-mining, non-

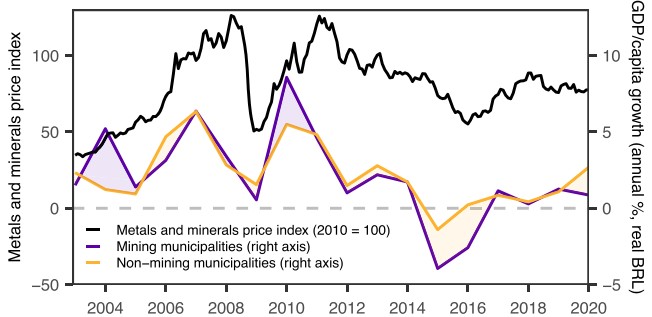

**Fig. 2 | Mining, commodity prices and the GDP.** Global prices of metals and minerals[82] and median annual growth rates of GDP per capita in real Brazilian Real[34,35,83] for mining and non-mining municipalities.

neighbouring municipalities, keeping all other municipality characteristics in the model constant.

The findings in Fig. 3a suggest that the local economic effects of Brazilian industrial mining varied over time, with notable differences between the periods before and after 2010. Prior to 2010, municipalities with industrial mining showed an average direct boost in GDP growth rates relative to non-mining lying between 1.6 pp and 2.1 pp (95% CI), alongside positive spillover effects of 1.3 to 2.5 pp (95% CI) in neighbouring municipalities (Fig. 3a, right panel). This spillover indicates that municipalities near those with industrial mines experienced additional economic growth of approximately 1.9 pp relative to municipalities without nearby mining activity. Since 2010, however, spillover effects have reversed, with municipalities near industrial mining experiencing an average GDP growth rate 0.5 percentage points (95% CI: −1.1 to −0.03) lower than that of municipalities without nearby mining activity. During the same period, the direct effect of industrial mining on GDP growth became statistically inconclusive, with estimates ranging between −0.1 pp and 0.3 pp (95% CI).

An analysis of yearly effect estimates (Fig. 3a, left panel) reveals that industrial mining consistently showed positive direct stimulus and spillover effects before 2010, aligning with the pooled estimates. The years since 2010 showed more variable results, with estimates ranging from positive to negative values in several years. Average yearly effects were generally lower than before 2010, often centred around negative values. This shift is particularly evident in the sharp decline observed after 2009: direct effects dropped from 1.5 pp (95% CI: 1 to 2.1) and spillover effects from 2.5 pp (95% CI: 1.4 to 3.5), to −0.8 in 2011 (95% CI: −1.4 to −0.3) and −1.2 pp in 2013 (95% CI: −2.3 to −0.04), respectively. Following this downturn, direct effects gradually recovered, returning to statistically significant positive values in 2014 and 2015. Spillover effects, however, remained statistically insignificant in the later years.

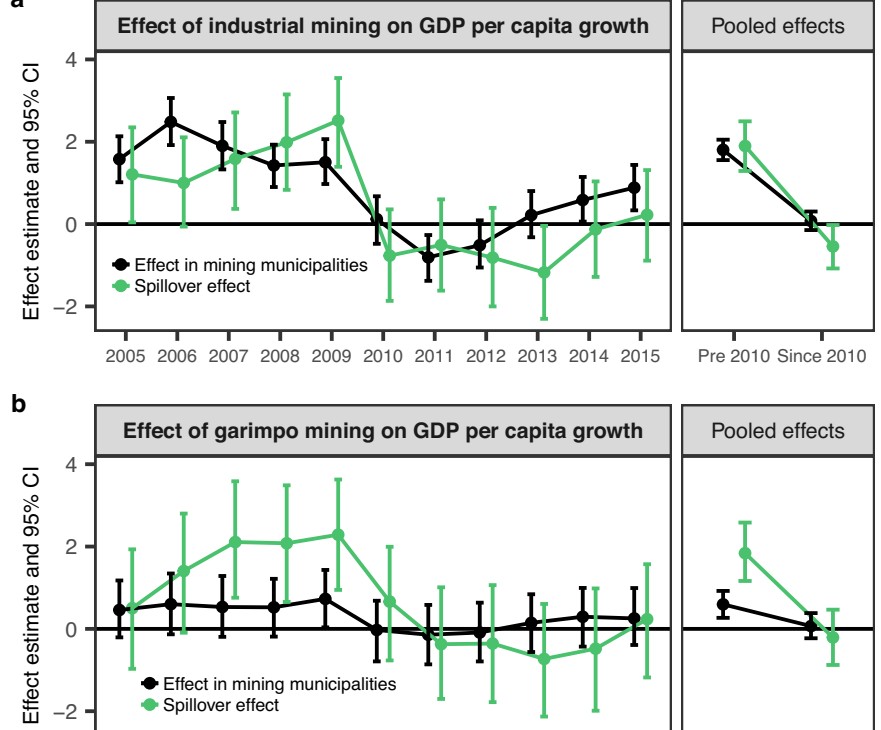

**Fig. 3 | Regional GDP effect estimates.** Relative effects of binary industrial (**a**) and garimpo (**b**) mining indicator on 5-year average annual GDP per capita growth, compared to control municipalities. Left panels show yearly estimates, right panels show pooled (pre-2010 and since 2010) estimates. Estimates were obtained from 20,000 Markov chain Monte Carlo iterations, with the first 10,000 being discarded as burn-in. Points denote posterior means, error bars show 95% posterior credible intervals.

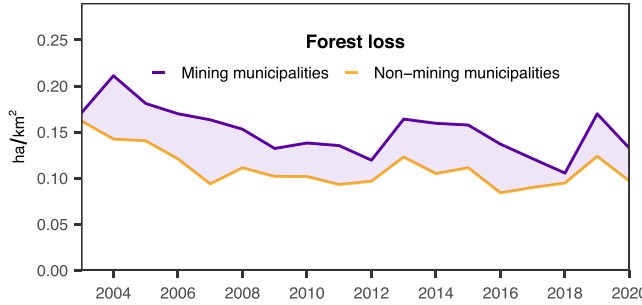

**Fig. 4 | Mining and forest loss.** Median annual forest loss rates (ha forest loss per km² municipality area, zero forest loss municipalities excluded) for mining and non-mining municipalities[18].

In contrast to industrial mining, garimpo mining exhibited weaker and less conclusive associations with GDP growth. For the pooled period before 2010, positive associations were observed for both direct (0.6 pp, 95% CI: 0.3 to 0.9) and spillover effects (1.8 pp, 95% CI: 1.2 to 2.6), though these were smaller than those linked to industrial mining (Fig. 3b, right panel). Annual direct effects were generally close to zero, with certainty intervals spanning both positive and negative values in all observed years, except for 2009, when a positive direct effect was identified (0.7 pp, 95% CI: 0.04 to 1.4). Unlike industrial mining, no positive associations were found for the most recent years of the sample. Spillover effect estimates were positive from 2007 to 2009 but disappeared thereafter. The full set of regression outputs, including all predictors, is available in the Supplementary Information.

### Mining and forest loss patterns

Figure 4 shows that Brazilian mining municipalities experienced higher forest loss rates than non-mining municipalities in the pre-matching data. Again, the figure merely suggests a positive link between mining and increased forest loss rates, however, without considering any hidden, potentially correlated deforestation drivers. We therefore followed the statistical approach introduced above, ensuring control over both observable and latent factors that could confound the relationship between mining and forest loss.

In contrast to GDP growth, where stronger effects were associated with industrial mining, forest loss was more closely linked to garimpo mining, particularly in the earlier years of the observation period (Fig. 5). Before 2010, municipalities with garimpo operations experienced, on average, 0.04 ha (95% CI: 0.03 to 0.06) more forest loss per km² compared to control regions. Spillover effects were more pronounced, with municipalities neighbouring garimpo areas losing an additional 0.29 ha (95% CI: 0.22 to 0.36) per km² of forest cover relative to those without nearby mining activity (Fig. 5b, right panel). In absolute terms, estimates suggest an average annual forest loss of 369 ha (95% CI: 311 to 426) directly within garimpo municipalities and an additional 1565 ha (95% CI: 1397 to 1725) per year due to spillover effects in surrounding municipalities (Fig. 5d, right panel).

Post-2010 data, as shown in the right panels of Fig. 5b, d, reveal weaker and less consistent associations between garimpo mining and forest loss. The relative forest loss estimate linked to garimpo operations diminished during this period. However, annual spillover effect estimates also suggest a re-emergence of an indirect effect in more recent years (Fig. 5b, left panel). Absolute forest loss effects also declined after 2010, with garimpo mining linked to an estimated annual loss of 84 ha (95% CI: 33 to 135) and a spillover effect of 301 ha (95% CI: 156 to 449) compared to non-mining controls (Fig. 5d).

In certain years, as well as in the pooled post-2010 estimates, municipalities with industrial mining activity appeared to have a protective effect on neighbouring forest cover, both in absolute and relative terms (Fig. 5a, c). However, this observed effect is not

supported by an alternative model specification that replaces binary mining indicators with ha of mining area (Fig. S5). Instead, the alternative model suggests that the forest loss spillovers from industrial mining are similar to those associated with garimpo mining. Within municipalities directly hosting industrial mines, the evidence remains inconclusive, suggesting that industrial mining may not have a consistent effect on forest loss at the municipal scale.

## Discussion

Our results suggest that fluctuations in global commodity prices not only contributed to economic stagnation and crisis in Brazil but also reshaped the observed relationship between mining and regional economic output, at times even reversing it. Before 2010, a favourable global economic environment, marked by high commodity prices and strong material demand, was associated with higher economic growth in mining municipalities and their neighbouring regions compared to similar non-mining areas. During this period, mining revenues and associated multiplier effects remained notably local. These findings contrast the negative relationship that is typically found in cross-country comparisons[28], instead indicating a positive association between mining and local GDP. This effect appears to transcend municipal borders, likely mediated by mechanisms such as labour market dynamics[10]. The observed positive spillovers point to the existence of diffused backward linkages, including the movement of commuting workers and the emergence of "mining clusters". Such clusters exhibit endogenous growth, fostering industrial diversification and strengthening the regional economy through agglomeration effects[25].

However, the expansion of the extractive sector can undermine other, potentially more sustainable economic activities, such as small-scale agriculture or manufacturing, while increasing dependence on mining[26]. Our results demonstrate that this dependence makes regional economies vulnerable to fluctuations in global commodity prices. When prices decline, the same backward linkages that once underpinned growth become pathways for economic contraction, affecting not only mining municipalities but also neighbouring regions. Over time, this volatility can undermine broader development efforts and reinforce economic instability.

Compared to industrial mining, garimpos showed weaker associations with economic growth, and we found no periods where effect estimates turned negative. One possible explanation lies in their frequently informal or illegal nature, which allows profits to evade official record-keeping or be transferred out of the region. At the same time, garimpo mining provides a livelihood for an estimated 200,000 people in the Brazilian Amazon[39], many of whom have limited economic opportunities. While GDP-based analyses may understate the local economic significance of garimpos, their broader socioeconomic role remains insufficiently understood. Future research should explore alternative, more granular measures of regional well-being to provide a more comprehensive assessment of garimpos' contributions to regional economies.

Stricter oversight of garimpo activities presents a promising opportunity to curb forest loss. Our findings reveal that the deforestation potential of mining strongly depends on the type of mining activities and the (in the Brazilian case dual) legal framework they are embedded in. This relationship is shaped, in part, by distinct geographical patterns: garimpos are overwhelmingly concentrated in the Brazilian Amazon, whereas industrial mining is more prevalent in regions such as Minas Gerais (see Supplementary Notes A). However, regulatory disparities also play a crucial role. Less stringent legal requirements – such as the absence of restrictions on mining techniques – opened the floodgates to adopting environmentally hazardous practices for holders of garimpo concessions. Over time, garimpos increasingly adopted methods and machinery of large-scale industrial exploration, deviating strongly from the original intent of the

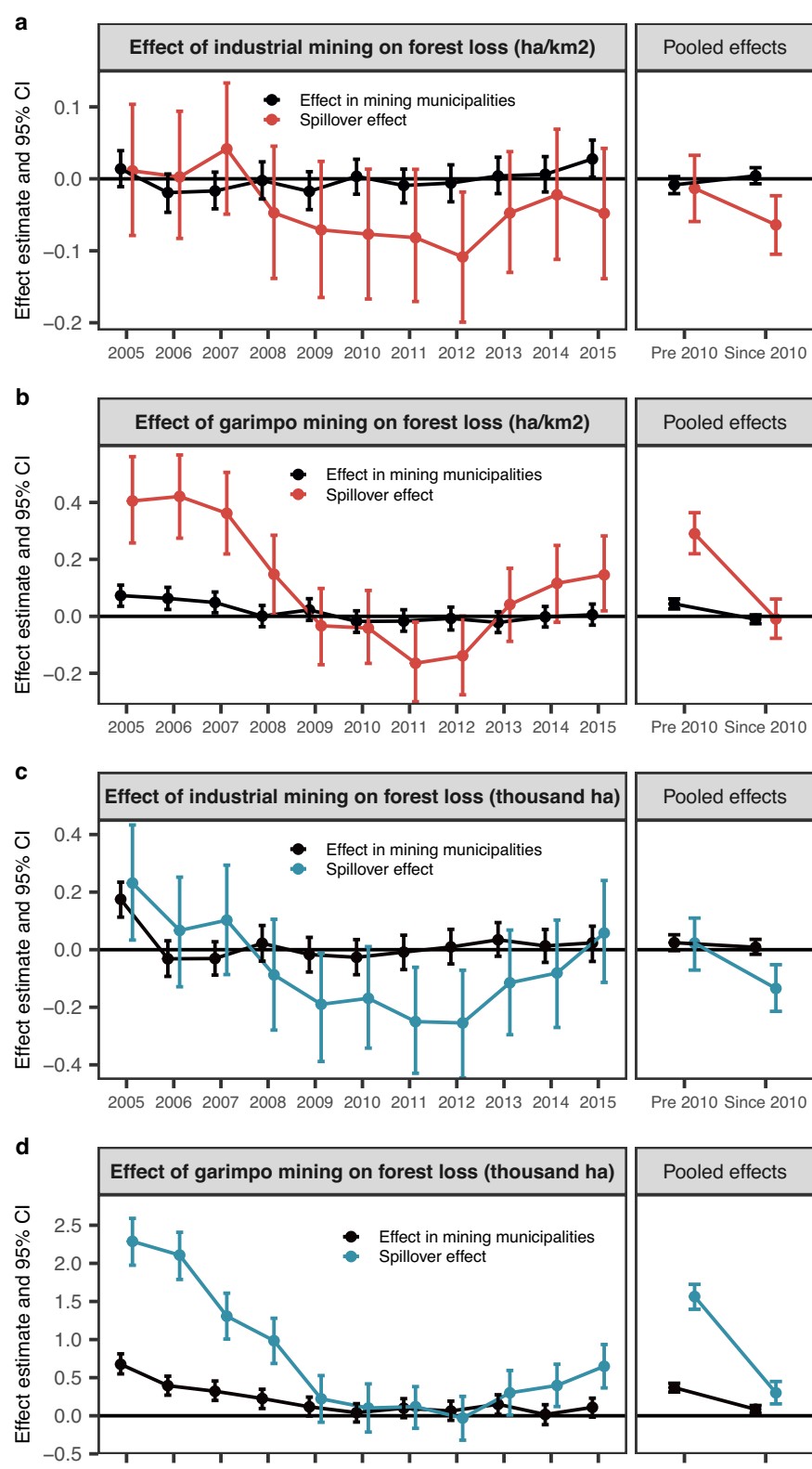

**Fig. 5 | Regional forest loss effect estimates.** Relative effects of binary industrial (**a**) and garimpo (**b**) mining indicator on forest loss in ha per km² of municipality area, compared to control municipalities. Relative effects of binary industrial (**c**) and garimpo (**d**) mining indicator on forest loss in thousand ha, compared to control municipalities. Left panels show yearly estimates, right panels show pooled (pre-2010 and since 2010) estimates. Estimates were obtained from 20,000 Markov chain Monte Carlo iterations, with the first 10,000 being discarded as burn-in. Points denote posterior means, error bars show 95% posterior credible intervals. Forest loss was defined as negative changes in natural forest formation based on ref. 18.

Garimpeira Mining Permission[40,41]. This trend provides a strong explanation for the higher rates of forest loss observed in garimpo regions compared to control areas. The pronounced spillover effects further suggest that garimpo mining is embedded within broader land-use transformations that contribute to deforestation. Rather than occurring in isolation, garimpos may accelerate existing processes such as logging, agriculture, and the expansion of infrastructure into previously undisturbed areas, reinforcing cycles of land degradation and resource exploitation. These findings highlight the urgent need for effective and enforceable policies to close regulatory loopholes, uphold environmental standards, and support improvements in the environmental performance of garimpos[42,43]. Formalising garimpos could play a key role in this effort, enabling better traceability of mined materials and ensuring compliance with environmental safeguards, such as mercury-free gold processing[44].

Finding a balance between forest loss and economic development is a central endeavour in making extractive economies and related supply chains more sustainable. Yet, our results revealed a more complex situation than a simple trade-off between the two spheres. While improved environmental responsibility must be achieved, especially in the poorly regulated informal sector, there is generally no guarantee for mining-induced economic development. While mining may benefit municipalities and their neighbours during boom phases, local mining economies can also experience bust phases associated with economic setbacks. This raises the question of which policies can mitigate negative economic impacts while retaining the positive economic stimulus during boom periods. Creating resilience to downturns is essential to guarantee the economic stability of mining operations. Long-term strategic planning by local authorities is needed in this regard, including far-sighted approaches to effectively use, allocate and invest mining rents for the benefit of local communities. Furthermore, these strategies should focus on diversifying local economies and reducing their reliance on the mining sector, thereby fostering economic resilience in the face of future uncertainties. Socioeconomic benefits could be strengthened through a fairer distribution of mining revenues – rather than the largest shares accruing to mining companies – and improved efficiency and transparency in the allocation of financial transfers to subnational and local governments[45]. Moreover, long-term planning is needed to address the socioeconomic prospects of communities in mined-out areas.

Revenues from taxes and royalties are considered contributing factors to the socioeconomic development of mining regions. The Brazilian CFEM (Financial Compensation for the Mineral Exploration) tax, for example, generates income for mining municipalities based on the volume and value of extracted material[10]. Moreover, local procurement and employment effects are frequently argued by the mining industry to foster regional economic development[46]. We showed that these economic dynamics strongly correlate with external factors such as world market prices. Municipalities may depend strongly on incomes and jobs from the mining industry, causing a reversal of the above-stated effects due to recoiling mining activities, job losses, diminished tax revenues, and a sluggish reorientation of local economic structures in times of falling market prices.

Our findings raise concerns about the alignment of mining with several of the United Nations' Sustainable Development Goals (SDGs). While the expansion of the mining sector is anticipated to provide essential resources for the global energy transition[47] with positive effects mainly on SDGs 7 (affordable and clean energy) and 13 (climate action), the regions tasked with this supply must cope with the multifaceted social and environmental challenges posed by mining. These challenges have been expected to be offset by the inherently inclusive and sustainable economic growth outlined in SDG 8[48]. Yet, our results indicate that relying on mining to realise this particular SDG may be misguided.

To conclude, our findings support the concerns raised earlier that the continued expansion of the extractive sector can increase deforestation and related pressures on Brazilian forests, particularly the Amazon, while deepening local economic dependence on mining[15,16,49,50]. This dependence heightens vulnerability to external shocks, such as fluctuations in mineral prices, and raises questions about the long-term stability and sustainability of regional development. The study thus contributes to a better-informed political debate by delivering much-needed quantitative evidence with relevance not only for Brazil but for resource-rich economies globally.

## Methods
### Econometric framework
The workflow is outlined in detail in Fig. S10 of the Supplementary Information. To evaluate the economic and environmental outcomes associated with the Brazilian mining sector, we employed a robust econometric framework widely applied to study the socioeconomic implications of resource extraction[33]. This methodology has also been utilised to explore key drivers of economic growth[51,52] and deforestation[53,54]. We applied a matching procedure to improve the balance between mining and non-mining municipalities, reducing statistical bias and mitigating model dependence[55]. By combining this balanced dataset with a comprehensive set of covariates in our regression design, we controlled for confounding factors – both observable and unobservable – that influence mining presence and its associated outcomes.

In light of the potential effects of mining activities on neighbouring areas, we aimed to account for the spatial spread of impacts across geographical locations. Combining well with the spatial nature of the mining data at hand, this study employed a spatial econometric approach. Spatial models explicitly consider the non-randomness of observations across space, thus addressing the bias and misleading inference that may result from spatial dependence[56]. Several applied contributions found strong evidence that socioeconomic and environmental observations were subject to spatial dependence at the regional level, including economic growth[51,57–59] and deforestation[54,60].

We employed panel-structure spatial models, accounting for spatial correlations and, at the same time, offering extended possibilities to consider time- or region-specific idiosyncratic effects[61]. Incorporating time-specific fixed effects, for example, accounts for factors influencing the dependent variables in specific years of the sample. The models can thereby consider trends affecting municipalities regardless of their exposure to mining, such as national (e.g., macroeconomic conditions, environmental and economic policies) and global (e.g., commodity price fluctuations) factors. We estimated the models using Bayesian methods following the standard Markov Chain Monte Carlo (MCMC) estimation framework as proposed for spatial econometrics[62]. The exact estimation procedure is presented in Supplementary Notes C. As a measure of uncertainty, we report 95% Bayesian credible intervals.

**Economic growth.** The underlying principle is to regress growth rates of countries or regions on income (usually GDP) at the initial period of a certain growth window as well as on a number of further determinants of growth. Typically, these include information on population growth, human capital stock and sectoral structure such as gross value added or employment across economic sectors[51,58].

Following the literature on economic growth and spatial spillover[52,58], and in line with the framework demonstrated for the Brazilian case[59], we employed a panel-structure spatial Durbin model (SDM) of the form:

$$\mathbf{y}_t = \rho \mathbf{W}\mathbf{y}_t + \mathbf{X}_t\boldsymbol{\beta} + \mathbf{W}\mathbf{X}_t\boldsymbol{\theta} + \boldsymbol{\xi}_t + \boldsymbol{\epsilon}_t, \quad \boldsymbol{\epsilon}_t \sim N(\mathbf{0}, \boldsymbol{\Omega}), \boldsymbol{\Omega} = \sigma^2 \mathbf{I}_n, \quad (1)$$

where $\mathbf{y}_t$ denotes an $n \times 1$ vector of regional economic growth rates at time $t$. As advocated in earlier literature[63], we used five-year periods as growth windows to smooth over short-term business cycle influences and calculated the respective average annual growth rates $\mathbf{y}_t = [\ln(\mathbf{Y}_{t+5}) - \ln(\mathbf{Y}_t)]/5 * 100$, with $\mathbf{Y}_t$ denoting per capita GDP at time $t$ (results were robust against variations in growth windows, see Fig. S7). $\mathbf{X}_t$ is an $n \times k$ matrix of $k$ exogenous municipality characteristics in the initial period. These include prominent determinants of economic growth such as initial income, population density, education and indicators for industrial structure, but also information on mining activities, land use and land use change. We used interaction terms between binary mining indicators and yearly dummy variables in order to obtain year-specific effects of the presence of industrial and garimpo mining. Additionally, in order to obtain pooled effects for the time before and after 2010, respectively, we ran another model interacting the binary mining indicators with dummy variables of the respective period. We selected 2010 as the separation point based on our yearly coefficient results, which revealed significant pattern shifts, including a marked change in industrial mining's GDP estimates and the conclusion of a period with particularly high absolute forest loss estimates for garimpos. The error term $\boldsymbol{\epsilon}_t$ was assumed to follow a multivariate Normal distribution with zero mean and a diagonal variance-covariance matrix $\boldsymbol{\Omega}$ with constant variance $\sigma^2$. $\mathbf{W}$ is an $n \times n$, non-negative, row-standardised spatial weights matrix. Its elements impose a structure of spatial dependence upon observational units, setting $w_{ii} = 0$ and $w_{ij} > 0$ if regions $i$ and $j$ are defined as neighbours ($i, j = 1, ..., n$). The exact specification of $\mathbf{W}$ is presented and illustrated in Supplementary Notes B and Fig. S4. Characteristically for an SDM, the regression equation includes the spatially-lagged dependent variable $\mathbf{Wy}_t$ as well as the spatially-lagged regional characteristics $\mathbf{WX}_t$ as explanatory variables. The $k \times 1$ vectors of the unknown parameters $\boldsymbol{\beta}$ and $\boldsymbol{\theta}$ correspond to $\mathbf{X}_t$ and $\mathbf{WX}_t$ respectively, and $\rho$ (where the sufficient stability condition $|\rho| < 1$ is satisfied for row-standardised $\mathbf{W}$[62]) is a scalar, measuring the magnitude of spatial autocorrelation. If $\rho = 0$, we obtain a growth regression model with spatial lags in $\mathbf{X}$ (SLX), where regional growth rates are independent, but $\mathbf{WX}$ is still considered. The model collapses into a classical linear model in the case where both $\rho = 0$ and $\boldsymbol{\theta} = 0$. Finally, the model considers a time-specific constant $\boldsymbol{\xi}_t$, capturing year-specific confounding factors such as commodity price dynamics and domestic business cycles.

**Forest loss.** This model type was designed to assess the effects of mining on forest loss, where again we used municipalities as observation units. Forest loss is expected to be subject to considerable spatial spillover[53,54,60], which is why we employed an SDM of the form

$$\tilde{\mathbf{y}}_t = \lambda \mathbf{W}\tilde{\mathbf{y}}_t + \tilde{\mathbf{X}}_t\boldsymbol{\delta} + \mathbf{W}\tilde{\mathbf{X}}_t\boldsymbol{\gamma} + \boldsymbol{\nu}_t + \boldsymbol{\mu}_t, \quad \boldsymbol{\mu}_t \sim N(\mathbf{0}, \tilde{\boldsymbol{\Omega}}), \tilde{\boldsymbol{\Omega}} = \tilde{\sigma}^2\mathbf{I}_n, \quad (2)$$

where the dependent variable $\tilde{\mathbf{y}}_t$ denotes a vector of cleared land within each municipality. In the $n \times \tilde{k}$ matrix $\tilde{\mathbf{X}}_t$ we considered economic growth directly as a control variable instead of including the full set of growth determinants. Other control variables remained the same as in the growth specification, because most determinants of economic growth overlap with indicators used for explaining forest loss[53]. Mining again entered the model in the form of interaction terms between binary mining indicators and year and period dummy variables, respectively. Similar to the growth model, $\boldsymbol{\nu}_t$ denotes a time-specific constant and we again assumed a normally distributed error term $\boldsymbol{\mu}_t$ with constant variance $\tilde{\sigma}^2$. The $k \times 1$ vectors $\boldsymbol{\delta}$ and $\boldsymbol{\gamma}$ correspond to $\tilde{\mathbf{X}}_t$ and $\mathbf{W}\tilde{\mathbf{X}}_t$ respectively and $\lambda$ is the spatial coefficient. The spatial weights matrix $\mathbf{W}$ and the properties of the spatial model remain the same as in Equation (1).

**Direct and spillover impacts.** Assuming independence of observations, the estimation coefficients of conventional (non-spatial) linear models can be typically interpreted as marginal changes in the dependent variable due to shifts in one of the explanatory variables. In this regard, spatial models require additional steps because we explicitly impose dependence among observations, implying that the partial derivatives of the dependent variable in region $i$ with respect to an explanatory variable in region $j$ are potentially non-zero and therefore cause feedback effects. Calculating average direct, indirect (i.e., spillover) and total impacts was proposed as a solution to this issue[62]: First, transforming Equation (1) (without loss of generality, the same derivation holds for Equation (2)) to

$$\mathbf{y}_t = (\mathbf{I}_n - \rho\mathbf{W})^{-1}(\mathbf{X}_t\boldsymbol{\beta} + \mathbf{WX}_t\boldsymbol{\theta} + \boldsymbol{\xi}_t + \boldsymbol{\epsilon}_t), \quad (3)$$

we derive $n^2$ partial derivatives of a particular explanatory variable $k$ as

$$\frac{\partial y_i}{\partial x_{jk}} = \mathbf{S}_k(\mathbf{W})_{ij} = (\mathbf{I}_n - \rho\mathbf{W})^{-1}(\mathbf{I}_n\beta_k + \mathbf{W}\theta_k)_{ij}, \quad (4)$$

where infinite feedback effects are captured through the spatial multiplier $(\mathbf{I}_n - \rho\mathbf{W})^{-1}$. The impact matrix is then summarised by calculating the average total effect as the average over all entries in $\mathbf{S}_k(\mathbf{W})_{ij}$, the average direct effect as the average when only considering its main diagonal, and the average indirect effect as the difference between the two. An interpretation of average direct effects is then given by the average response of the dependent to the independent variables over the sample of observations and hence similar to regression coefficients from classical linear models. The average spillover can be interpreted as the cumulative average response of a region's dependent variable to a marginal change in an explanatory characteristic across all other regions.

### Data
We compiled a balanced panel dataset covering 5262 Brazilian municipalities over the period 2005–2020. Calculating five-year average growth rates, this resulted in 57,882 observations prior to matching. Data were sourced from multiple databases and, where necessary, aggregated to the municipality level. Municipalities, the smallest administrative divisions in Brazil, occasionally undergo boundary changes due to splitting or merging, resulting in a variable total count over time. In order to keep a balanced panel with a constant number of spatial observations, we followed previous research[59] and only considered municipalities with unchanged geographical extent over the sample period. Table S1 provides an overview of the variables used in this analysis.

**Dependent variables.** The dependent variable in the growth models was the five-year average annual growth rate of GDP per capita, which was computed from yearly per capita GDP in BRL at current purchasing power parities as reported by the Brazilian Institute for Geography and Statistics, IBGE[34,35]. In the last year of the panel, 2015, this measure therefore comprises economic growth between 2015 and 2020. We selected five-year growth windows as a suitable measure for mid-term economic effects[63]. We were aware that other studies emphasise poverty and distributional effects of mining[45,64]. However, we needed to resort to GDP growth in our study because alternative socioeconomic indicators that would allow for a broader understanding of human well-being are difficult to obtain for this level of geographical detail, especially for a yearly panel. GDP per capita, therefore, served as a key indicator for economic development, despite its limitations, such as only covering market transactions and not describing income distribution.

The municipalities included in the analysis varied in size, ranging from 3.57 km² to 159,533 km² (mean = 1541 km², sd = 5683 km²). For the forest loss models, we used two measures of the dependent variable to capture the reduction in natural forest formation: (a) annual relative change, expressed in ha per km², and (b) annual change in absolute ha. The data was calculated from municipality-level land cover statistics as provided by the MapBiomas project[18]. In contrast to the five-year windows used in the economic growth specification, forest cover changes were examined annually. This decision reflected the assumption that deforestation is typically driven by immediate events, with no meaningful recovery periods, making adjustments for business cycles unnecessary.

**Mining indicators.** The essential municipality characteristic for this study was the presence of mining activities. Mining entered the models as binary indicators for the presence of industrial and garimpo mining within a municipality in a certain year. Detailed yearly geospatial data on Brazilian land area covered by mining was taken from MapBiomas[18]. We transformed their continuous metric (ha per municipality) to a binary variable for simpler interpretation of effects. In the Supplementary Information, we show that alternative model specifications, which incorporate the mining area in ha, do not alter our main conclusions (Figs. S5 and S6).

Instead of land cover information about mining, the CFEM tax would have been another indicator for active mining activities. However, we refrained from using the CFEM as an explanatory variable, as the tax income directly enters municipality GDP, i.e., the dependent variable, creating identification issues in the econometric model.

**Covariates and potential confounding factors.** We accounted for a broad set of covariates in the design matrices $\mathbf{X}$ and $\tilde{\mathbf{X}}$ to better isolate the effect of mining on GDP and forest cover by controlling for confounding factors that could simultaneously influence these outcomes and the presence of mining. By addressing such confounding variables, we sought to reduce bias and strengthen the robustness of our findings. The selected covariates reflect key local environmental, economic, and sociodemographic conditions, which are further detailed in the text. A comprehensive list of these covariates, along with the rationale for their inclusion, is provided in Tables S2 and S3.

We considered land use change dynamics as control variables in all models. Using satellite data on the conversion of land, e.g., from natural forest formation to pasture or from grassland to agriculture, is an efficient approach for observing economic activity and environmental transformation at the same time. Such patterns act as useful proxies for underlying factors, including fertile soils, residential development, or conservation, which may confound our analysis by simultaneously influencing mine development (e.g., fertile soils may attract land uses that compete with mining) and the outcome variables of our interest. Our data was obtained from MapBiomas[18], providing yearly land transition information from 30 m resolution satellite images aggregated to the level of Brazilian municipalities. We utilised land cover classifications at the second sub-categorical level and considered forest formation and forest plantation for the case of forest, grassland as non-forest natural formation, and agriculture and pasture for farming. Other categories such as wetlands, non-vegetated areas and bodies of water were omitted since they had minor relevance for our analysis. In order to be consistent with the five-year GDP growth horizon, we computed the average change in ha over five years. Land use change from any category to forest formation was not considered as a covariate, because it marks a transformation that is only viable over a longer time horizon.

Initial land cover was considered as a proxy for the land cover conditions at the beginning of either a window of GDP growth or a one-year forest loss period. We again used data from MapBiomas[18]. In order to reflect the variation in municipality area, this variable entered the

models as shares of natural forest, forest plantation, grassland, agriculture and pasture relative to the total municipality area.

The remaining covariates were motivated by economic growth theory (see the respective literature below), and by following a meta-analysis for the case of the forest loss models[53]. While some of these variables are not obvious confounders – i.e., they may influence the dependent variables without affecting mining expansion, or they may only exert an indirect influence via other channels – their inclusion is expected to improve the precision of effect estimates by accounting for additional variation in the outcomes. We considered initial income in terms of per capita GDP in the initial year of a growth window as a proxy for physical capital, which is a major determinant of economic growth in the neoclassical growth framework[65]. A negative relationship between the initial stock of physical capital and economic growth, which is explained by diminishing returns to capital accumulation, is a well-established stylised fact in the empirical literature known as the convergence hypothesis. In addition, a number of studies show that the convergence hypothesis holds for direct impacts in spatial econometric growth frameworks, while spillover effects from the flows of capital, goods, knowledge, and people between regions are shown to be positive, implying that relatively poorer regions benefit from having highly capitalised neighbours[57].

Endogenous growth theory highlights the role of human capital as a key driver of innovation processes such as technological change[66,67]. However, whether indirect effects are positive or negative is uncertain because positive economic effects from knowledge spillover and brain drain channels may counteract each other. The role of the labour market in shaping mining activity is similarly ambiguous and likely context-dependent. Higher levels of human capital may attract industrial mining by reducing training costs and increasing productivity, while simultaneously discouraging informal or artisanal mining, as more educated populations are likely to pursue alternative livelihoods. We proxied human capital using the FIRJAN education index[36], an index for Brazilian municipalities on a scale from 0 (worst) to 1 (best), measuring both schooling coverage and quality. The education index was only available from 2005, constraining our sample to this starting year.

Population growth was another component taken from the neoclassical growth framework. Following this theory, a positive impact of population growth would hold for absolute income growth at the national scale, but not for the growth of per capita income due to capital dilution. Therefore, unless higher output exceeds population growth, we would expect a negative effect. For subnational entities, this relationship is unclear, because one part of the population dynamics is migration patterns, which may vary across scale levels[59]. We obtained population counts per municipality from the IBGE and computed population growth again at five-year average rates. Population counts for 2007 and 2010 were interpolated due to missing data.

In line with numerous other studies, we used population density as a proxy for agglomeration externalities[59]. Population agglomeration effects have been considered in the economic geography literature. Denser populated (i.e., urban) regions are associated with positive effects on productivity growth, because they show higher rates of technological progress[68]. However, this relationship may not hold for relatively poor districts in countries where strong urbanisation is driven by extensive population growth without substantially affecting labour productivity.

We followed previous research[58] and included the gross value added (GVA) in the agriculture, industry and service sectors as control variables in order to proxy the industrial structure of municipalities. These variables help account for heterogeneities in economic growth and forest cover dynamics, as well as underlying processes such as land competition and local material demand, which may either facilitate or constrain the development of mining activities.

The forest loss regressions followed a similar structure as the growth regression, with small adjustments. Instead of initial income, population growth and density, human capital and the sectoral mix variables, we directly used the five-year average annual economic growth rates as a proxy for economic activity. The empirical literature is inconclusive regarding the direction of the effect that income may have on forest cover[53], yet this proxy subsumes a set of deforestation drivers that are related to any other anthropogenic activity besides the mining and other land use change effects. Forest cover change accounts were only included in the GDP growth models, as they effectively define the dependent variables in the forest loss models.

Precipitation and elevation were included as final control variables for economic growth and forest loss. These biophysical characteristics capture key spatial constraints that may influence GDP growth and forest cover, as well as factors relevant to mining presence or expansion, such as accessibility, land development potential, and susceptibility to wildfires or erosion. The data was compiled using the `dplyr`[69], `tidyr`[70], `readxl`[71], `stringi`[72], `sf`[73], `raster`[74], `geobr`[75], `exactextractr`[76] and `elevatr`[77] R[78] packages.

While we have carefully accounted for a comprehensive set of covariates, it is important to acknowledge that residual confounding may still persist. Unobserved or complex factors, which are difficult to capture in the model, could influence the relationships between mining, GDP, and forest cover. Nevertheless, the inclusion of time-fixed effects and the longitudinal panel design that tracks outcomes over time, helps mitigate the potential for such unobserved heterogeneity[61]. Moreover, the large sample size and temporal coverage of the data add confidence in the reliability of our estimates.

**Statistical matching.** To mitigate confounding and improve the robustness of our estimates, we employed coarsened exact matching (CEM), which approximates randomisation by grouping treated and control units based on similar characteristics[55]. CEM offers several advantages over other matching techniques, including its ability to handle high-dimensional covariates without relying on strong parametric assumptions. It also provides a better balance between the treated and control groups compared to other methods. The matching was implemented in R using the `cem` package[79].

We applied a one-to-many matching approach using variables population density, natural forest cover, precipitation, elevation and municipality area to account for factors influencing the development of mining while preserving a sufficiently large sample. Matching was conducted separately for industrial and garimpo mining municipalities. As a result, the number of industrial mining observations was reduced from 4430 to 3923, while garimpo mining observations decreased from 2753 to 2356. The matched control groups comprised 42,646 and 35,156 observations for industrial and garimpo mining, respectively (Table S4). After further refinement to ensure a balanced panel, the final samples consisted of 33,836 and 23,727 observations for industrial and garimpo mining, respectively, as reported in the regression outputs in Tables S5–S16.

While the matching procedure enhanced balance between treated and control groups by excluding cases poorly suited for comparison, some differences remained – most notably in forest cover, where certain mining municipalities exhibited higher initial levels. This residual imbalance is addressed in the econometric models by incorporating initial forest cover as a covariate. The remaining imbalance is partly due to the necessity of retaining spatial information. Unlike studies that apply one-to-one matching[32], we chose a broader approach to preserve neighbouring municipalities and the overall spatial structure. Our matching strategy also follows recommendations for CEM[80], prioritising improved balance while maintaining a sufficiently large sample. Despite these limitations, the matching procedure enhanced the comparability between groups, reducing bias and strengthening the reliability of our findings, as shown in Fig. S11.

**Reporting summary**

Further information on research design is available in the Nature Portfolio Reporting Summary linked to this article.

## Data availability

The compiled dataset used in this study, along with all materials required to reproduce the results, has been deposited on Zenodo and is available at https://doi.org/10.5281/zenodo.10009935. A detailed overview of the variables and their original sources is provided in Table S1 of the Supplementary Information. Step-by-step instructions for accessing and working with the data are available at https://github.com/SLuckeneder/mining-Brazil.

## Code availability

All code used in the analysis is available at https://doi.org/10.5281/zenodo.10009935.

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

## Acknowledgements

The authors thank Anthony Bebbington, Jesus Crespo Cuaresma, Stefan Giljum, Richard van der Hoff and Tim Werner for valuable comments and discussions. This work was funded by the European Commission under the ERC Consolidator Grant FINEPRINT (Grant No. 725525; SL, VM) and by the Austrian National Bank (OeNB) under Grant No. 18799 (SL). Part of the research was developed in the Young Scientists Summer Programme at the International Institute for Applied Systems Analysis, Laxenburg (Austria) with financial support from the Austrian IIASA Committee (SL). The work was furthermore supported by the Austrian Federal Ministry of Education, Science and Research (Marietta Blau Grant MPC-2022-02358), which was managed by the OeAD-GmbH (SL).

## Author contributions

S.L. conceptualised and conducted the study. V.M. and T.K. contributed to the code development. J.S.G. and M.K. provided intellectual input on various aspects of the study. S.L. wrote the first draft of the manuscript, and all authors contributed to subsequent revisions.

## Competing interests

The authors declare no competing interests.
