## [Transparent Peer Review file · Nature Communications]

Forest loss and uncertain economic gains from industrial and garimpo mining in Brazilian municipalities

Corresponding Author: Mr Sebastian Luckeneder

Version 0:

Reviewer comments:

Reviewer #1

(Remarks to the Author)

This is an interesting work. I have a few suggestions for further improvement:

1. The paper uses dummies for modeling the presence or absence of mining activity. If there is data, one could also use actual mining intensity to see if that explains the effects of mining on GDP and forest loss.
2. The model is concerned with regional GDP growth. However, it is obvious that mining activities rarely benefit the local regions. Is it possible for the authors to also model the effect of mining at a national level to get a more comprehensive impact of its positive effects?
3. The effect of mining on deforestation is clear. However, does deforestation also provide a short-term positive economic benefit and perhaps a long-term negative benefit? Is this worthwhile further investigating?

Reviewer #2

(Remarks to the Author)

This is an interesting and well-written manuscript. The authors claim to show mining's impact on deforestation and regional economic growth in Brazil. The main findings reported are that: a) Mining seems to impact GDP in the mined municipalities and their neighbours - but that this is linked to the commodity price, and b) less well regulated mining increases deforestation.

However, I am not sure the data supports the claims made in the text. As an example, I will focus on the two key figures - Figure 2 and Figure 3. Figure 2a shows very few differences between the GDP trends in mining and non-mining municipalities - the two track each other nearly perfectly, implying mining has very little impact on local GDP (but it could well be driving these patterns at a wider [e.g. national] scale - i.e. responsible for the trends in mining and non-mining states due to income from mining being spent at country-level). Figure 2b is also concerning. Focussing on the 95% estimates shows very strong overlap both before and after 2010 between mined municipalities and their neighbours - in essence they are showing near identical trends. The authors claim this trend is due to mining but, since non-mined, non-neighbour municipalities are not shown on this graph, there is no evidence that other non-mined states show different trends. Indeed, Figure 2a suggests the opposite (i.e. that the non-mined states show the same trend). If the trends are the same in the mined and non-mined municipalities then it is highly unlikely to be driven by the mining! Similar statements can be made about Figure 2c and about Figures 3a, 3b and 3c. However, as the differences in trend between mined and non-mined municipalities seem greater (Figure 3a), then there is a potential impact of mining here.

I am not an expert in the methods the authors have applied, and so I encourage the editors to seek out a methodological expert in impact evaluation. However, whilst I am not convinced the mined vs non-mined municipal comparison shows any meaningful difference (see above), I think the comparison is oversimplistic. Robust impact evaluation requires a proper counterfactual and it is unlikely that the non-mined municipalities provide this. The methods would be vastly improved by using statistical matching to show that some of the non-mined municipalities provide a good counterfactual or, indeed, if they do not, then perhaps constructing a synthetic control. A BACI is the gold standard in impact evaluation and should be performed here as it would allow for other potential differences to be accounted for (e.g. wider changes in GDP driving by national policy that impact the mined municipalities and the counterfactual identically). I suspect that once these have been

accounted for, there will be no difference between the GDP in mined municipalities and their neighbours, nor with non-mined municipalities either! If so, this would lead to a very different conclusion to that drawn by the authors.

Lastly, due to a lack of line numbers in the manuscript, it is very difficult to give feedback on more minor errors within the manuscript.

Version 1:

Reviewer comments:

Reviewer #1

(Remarks to the Author)

The authors have responded to my comments; however, I still believe that the impact of mining on deforestation has not been convincingly addressed. The paper finds a substantial impact of mining on deforestation. Therefore, the logical next step would be to ask whether the economic impact of this deforestation has been positive or negative, which the authors have not addressed in the revised version.

If deforestation is not at all a channel through which the economy is impacted, why dedicate a major section of the paper to it? If the paper aimed solely to estimate the impact of mining on deforestation, it would make sense to leave it as it is. However, since the goal is to highlight the impact of mining on the local economy and deforestation, it would make sense to test whether areas with deforestation have seen any adverse impacts on their agricultural GDP, natural resources, etc.

At the moment, the paper appears somewhat disjointed, with one aspect exploring the impact of mining on the economy and the other exploring the impact on deforestation. Spending some effort on linking the two would benefit the paper and better inform the readers, in my opinion.

(Remarks on code availability)

Reviewer #2

(Remarks to the Author)

Thank you for these revisions - they have improved the manuscript and make it much clearer.

I still have some concerns relating to the method used - namely that before-after control-impact assessments would be a more appropriate method to use. As the authors acknowledge in their response letter, our study designed are more appropriate for making strong causal claims. The authors indicate that they therefore refrain from these claims. However, I do not think that is the case and, as such, without amendments, the manuscript is misleading - i.e. makes claims that are not supported by the results due to the methods used.

For example, referring to the line numbers of the version with tracked changes...

Line 2: The title is misleading. All you have shown is correlation between transient economic benefit, forest loss and mining. You have not identified causation and so you do not know if these are the regional impacts of mining or not.

Line 10: You don't investigate the impact of mining on deforestation and regional economic growth - you investigate the correlation of these things.

Line 11: "Substantial effects of mining on deforestation rates" implies causation, you just show the two are correlated.

Lines 70-71: The authors acknowledge that their approach contrasts with before-after control-impact assessments and what the latter requires. However, this does not go far enough. The authors need to explain the advantages and disadvantages of the method they have chosen and the limitations associated with it (i.e. that they identify correlation and not causation).

Line 119: the findings do not reveal the local economic effects of industrial mining activities. They reveal the correlation between local economies and mining activities.

Line 122: Again, you mention the impacts of mining, but you only show correlation.

Line 130: Again you mention the effect of mining, but have not shown causation.

Line 154: This heading is misleading - the forest loss may not be mining-induced, you only show correlation not causation.

Line 190: 'translated into' implies causation, which is not supported by this method.

Line 214: Again, this implies causation.

This is only a small sample of where the language used by the authors potentially implies causation and so goes beyond what their results are able to support. The text throughout the manuscript needs editing to ensure language such as that highlighted above are not used, and the manuscript very clearly refers to correlations and not 'impact'.

(Remarks on code availability)

Reviewer #3

(Remarks to the Author)

This paper is on a very important topic (the impacts of mining on deforestation and local economy), I was therefore very interested to read this paper. The paper is well written and the applied problem is well positioned in the literature. The data and code are available which is great to see. This is clearly a resubmission and I read the thoughtful response to reviewers the authors had prepared. This was all good. They made appropriate changes. However, despite all of this, I hesitate to recommend this paper for publication in Nature Communications. While in the response to reviewers they respond to the query about the methods they use to draw causal conclusions by claiming not to make strong causal claims in the paper, the paper is framed as exploring and demonstrating causal relationships. I don't see how they could downplay this-if they are simply demonstrating an association, then the paper is not interesting. However, I am not convinced that the study design is sufficient to base causal claims on and I do not feel that they sufficiently acknowledge the assumptions underlying their identification strategy (or provide support for these assumptions). However, these comments come with a warning that I do not completely follow their approach (I have not used the sort of spatial model they use myself). I would encourage the editor to approach something with very specific technical expertise in evaluating causal claims from this sort of spatial model. I am afraid I don't know who that would be.

Given the caveat that I am not an expert in the precise methods used, I add some reflections.

Causal claims in this context (which they do make, despite their comment in the response to reviewers) depend on the assumption that once they have controlled for observed confounders, that allocation of municipalities to the mining or no mining group is as if random. It is very difficult to believe this assumption is true especially given that they do not appear to do any matching to ensure the set of municipalities included in the analysis is at all similar. One concern I had was that they are presumably including in the analysis both municipalities with almost no forest cover and ones with lots of forest cover. A low loss of forest in terms of hectares per municipality area might be because the municipality has little forest, or because little has been lost. If I have missed that their model accounts for this I am sorry (it is possible but I am not clear where). I would have really liked to see a table showing the confounding variables their design takes account of (and the datasets used to capture them). A DAG would also be helpful.

In any analysis such as this, the question is always whether there are likely to be hidden confounders which are large enough to disrupt the results. I can think of plenty potential confounders. A solid hidden confounder analysis would be valuable (though I am not sure how it can be implemented with this model, as I say I am not familiar with it).

Finally, I find the claim that the design shows spill over rather unconvincing. Could not the same hidden confounders drive patterns seen?

Smaller points:

Figure 1 is really helpful. It presents really useful data and helps us understand its structure. Tiny point: you could label the regions more clearly, I struggled with them.

Y axis isn't labelled in Fig 4

(Remarks on code availability)

I looked at the code and checked the data was there but no more.

Reviewer #4

(Remarks to the Author)

Thank you for this submission. It is a really interesting study and I enjoyed reading it.

I have three main areas of concern which currently prevent me from recommending that this article be accepted. However, they mostly relate to the communication of results and conclusions and therefore should be a relatively easy fix:

1) I agree with some of the concerns of Reviewer 2. Without a path diagram/theory of change outlining your assumptions I am not certain that you have controlled for all possible observable confounders of the causal relationship between mining GDP growth/forest cover change. I agree with the other reviewer that there could be outstanding differences between municipalities with and without mining which could confound your estimates. For instance, perhaps mining municipalities have specific geological characteristics which affect soil productivity and agricultural development, and consequently GDP and forest cover. Having looked at Supplementary Table 1 I think that many of your variables probably do capture or proxy for many possible confounders but as authors with much greater knowledge of the study area than me, I think it's your responsibility to make this crystal clear. So, I would like to see an explanation (probably in SI due to word limits and briefly outlined in the main text) of the potential confounders, and how your data capture or proxy for these. If you think there is a reasonable chance that there remain unobserved confounders, you could include a statement that your results reflect association between the presence of mining and outcomes but not causality.

Also, is there any reason why you haven't used unit-fixed effects to control for time-invariant heterogeneity?

Are you sure that you have controlled for factors that may uniquely affect GDP growth rate in the control municipalities but not the mining ones?

Finally, for the benefit of the editor, I am happy that the time fixed effects do control for the contribution of national-scale factors affecting all municipalities to the GDP growth rate.

2) I note that in your previous rebuttal letter you say that you have been careful about making causal claims and I think this is good, given that there may still be unobserved confounders. However, I disagree with some of the reporting of your results. I

am not convinced that your results support the language you use to report your main findings. In the more detailed points below I have highlighted instances where I think your language is too strong.

For example: You show that municipalities with industrial mining are associated with lower economic growth than those without for 3 years (2010, 2011 and 2012). The number of years with a negative effect is exceeded by those with a positive effect, and the maximum positive effect is higher than the maximum negative one. Yet, in the introduction you state that 'the associated boost in GDP is transient', however, I would argue that the associated drop in GDP growth is transient as you show that the positive effect of mining recovers from 2014.

No-one ever likes to tone down the strength of their main message but I think that with the suggested amendments your main message still stands – that the economic contributions from mining are not guaranteed given they are dependent on global commodity prices and therefore vulnerable to shocks. However, you need to make sure your reporting aligns with your results.

3) I think a confounding factor could be affecting the estimates of spillover of deforestation from garimpo mining, because I am struggling to believe the magnitude of effects. I can think of processes through which the spillover is working in the opposite direction – a deforestation frontier develops in neighbouring municipalities, more people arrive and spread into adjacent municipalities where they then discover gold, leading to the expansion of deforestation for garimpo mining. In this case the potential causal relationship goes the other way; increasing deforestation in neighbours is a proxy for frontier development and in-migration which then causes the discovery of gold in the mined municipality.

In the other direction, indirect deforestation associated with mining is usually concentrated relatively close to the mining area (e.g. within 5km: <https://www.nature.com/articles/s41893-024-01421-8>). Unless all the mining areas are very close to municipality borders, I'm not convinced that this cause enough spillover deforestation to cause such a large effect. I suppose the frontier effect could be at play and people attracted by mining could cross municipality borders and clear land for farming/pasture, but in that case I find it strange that mining wouldn't also expand. Particularly given that the gold exploited by garimpo miners are usually secondary deposits along river beds which can extend long distances, crossing municipality boundaries.

If you believe this pattern of spillovers detected in your results does reflect the reality in Brazil, please explain this in text and the potential mechanisms behind this effect. If you think it might be spurious, please also discuss this.

Question:

- Are you sure that the non-mining municipalities never mined?

Specific points:

1) Lines 10 – 15: See comment above. I disagree with this statement. Given that the positive effect of industrial mining on GDP growth recovers in 2014 and 2015, I think the negative effect is transient. I recommend you talk instead about the effect of industrial mining on GDP growth being inconsistent and unreliable, as it is vulnerable to global mineral prices. Based on the evidence presented here I don't think you can say that 'Brazil's extractive industries have failed to deliver lasting economic advantages'. If looked at differences over the whole time period 2005 – 2015 you might find that mining did have an overall positive effect, or at least not a negative one. So please delete this sentence. However, you can ask questions about the sustainability and reliability of mining-based economic growth.

I also think you should separate the sentence on the effects of garimpo mining on deforestation from the sentence about the effects of industrial mining on GDP growth. Otherwise it sounds like you are talking about the same type of mining when you are not.

2) Line 11: I also wonder if you should say 'less regulated garimpo mining concessions' for those who are familiar with small-scale mining (although the editors might disagree!) Or 'less regulated small-scale mining concessions'.

3) Line 3: I think you need to mention governance when talking about the resource curse because this is the main reason why in some countries, resource wealth has not translated to GDP growth and poverty reduction. Lack of investment in health, education and manufacturing is a symptom of that.

4) Line 55: Use the term 'artisanal and small-scale mining' here as this is a key term for this sort of mining.

5) Line 102: replace 'impact' with 'effect'. Also, Fig. 3 caption says that you are plotting the 'posterior means', yet in the text it says median? Then below in lines 107 you say average too. I think you are plotting the average difference, so please change 'median' to 'average'.

6) Line 101: 'Ambivalent' is not the correct term to describe this pattern (I don't think a positive effect in 7 years compared to 2 with 2 with a negative effect qualifies as ambivalent). Please use 'varied' or 'variable over time'. Don't say 'across Brazils 5262 municipalities' because it sounds like the effect varied over space, rather than time.

7) Line 124: Change 'uncertainty' to 'estimates'

8) Lines 152 – 154: Unnecessary and confusing, please delete

9) I guess those estimates of 894 ha per year of direct deforestation from garimpo is the estimated total across all mined municipalities? I am struggling to believe the magnitude of spillover effects and the mechanism behind it here. Usually indirect deforestation associated with artisanal and small-scale mining is relatively close to the mined area (i.e. within 5km, see (<https://www.nature.com/articles/s41893-024-01421-8>). See main point #3 above. Please add an explanation if you believe this is a real effect or potentially spurious.

10) Lines 167 – 169. This is very interesting!

11) Line 174: please don't use the term 'causing' because you have shown association, not causality.

12) Lines 194 – 196: Please amend this statement. Some garimpo profits will feed back into the local economy through local spending by miners, traders and those further down the supply chain. I totally agree that the fact that the gold is traded informally probably contributes to the evidence of no effect on GDP, but I don't think you say that no profits are spent locally.

13) Lines 213 – 215: If you are going to talk about banning garimpo gold mining then you need to say that this will have serious impacts on the livelihoods of miners, many (but not all) of whom I guess are very poor. Perhaps a better suggestion would be to focus on the need for better regulation and support to improve environmental practices (there is lots of literature

on this from this from which you can draw examples). Also because there is heaps of evidence from around the world that bans just don't work, and often end up making things worse by pushing miners into more remote areas and increasing conflict.

14) Lines 223: 'stronger negative impacts' than positive impacts is not supported by your data, please delete.

15) Lines 225 – 227: 'long-term downturns (as discussed in the resource curse literature'. Long-term downturns sounds like it is related to economic crises/commodity price drops and I don't think the resource curse literature particularly focusses on this (but I could be wrong). I think it's best to just delete this and say 'creating resilience to downturns is essential to guarantee the economic stability of mining operations'.

16) Lines 230 – 231: I think that this statement contradicts your findings. Apart from the 4 years where mining has a negative or zero effect, for all the other years you show that industrial mining had a positive effect on GDP growth. And you state that local economies 'constitute the concrete living environment of the local population' - so do benefit citizens. Please rewrite this sentence to be more specific. Perhaps you could say that the positive socio-economic effects could be greater with a more equal distribution of economic benefits, and that this could help to buffer the effect on local economies during shocks.

17) Line 257: 'while likely failing economic promises' I think this is too strong. Please tone down. You could say something like 'while making local economies dependent on mining. This increasing vulnerability to external shocks (e.g. in mineral prices), and raises questions about the sustainability of economic development.

18) Lines 315 – 316: This sentence could do with more explanation.

19) Lines 470 – 473: I'm not sure that the average GDP growth rate would capture all other anthropogenic drivers of land use change. What about smallholder agriculture?

-

(Remarks on code availability)

Version 2:

Reviewer comments:

Reviewer #1

(Remarks to the Author)

(Remarks on code availability)

Reviewer #2

(Remarks to the Author)

The authors seem to have addressed my concerns. I have no further comments

(Remarks on code availability)

Reviewer #4

(Remarks to the Author)

Uncertain economic gains and forest loss: Evidence from industrial and garimpo mining in Brazilian municipalities. Review 3

Main points:

Thank for your kind and thoughtful response to my last comments and the substantial changes you have made to the manuscript. It is much improved now. Thank you for changing the title too - it is much better aligned with your results now. Can I just check though - do you intend to say that the forest loss from mining is uncertain? If not, consider switching the title to read: 'Forest loss and uncertain economic gains:...'

Finally, I want to thank you for including garimpos in your study and for amending the text to add the nuance I recommended. The impacts of artisanal and small-scale mining has been so under-researched and when it has been studied the discussion has often been too one-sided. So, thank you :)

I still think how you present the confounding factors could be improved. In the section Covariates and potential confounding factors you haven't explicitly highlighted which variables you think are correlated with the presence of mining and causally effect GDP and forest loss (i.e. the confounding factors). This is made more confusing by the fact that in Supplementary Tables 2 and 3 it is only discussed how the variables influence outcomes, not treatment at all. In causal inference studies

aiming to ascertain the effect of an intervention you only need to control for, via matching and regressions, hypothesized confounding factors. If you think precipitation only influences forest loss and not the presence of mining, you don't need to control for it because you are saying it is okay to assume that mining is randomly allocated along the precipitation gradient, therefore precipitation will not affect average differences in outcomes between treated and control units.

However, in complex land systems sometimes while a factor might not directly affect allocation to treatment it could be correlated with something else that does, meaning allocation to treatment is not as if random across this gradient. For example, I can see in Figure S11 that there were differences in precipitation between mining and non-mining mining municipalities (although still relatively small). For that reason, I think it's probably okay to include non-directly confounding factors in the regressions, just in case (although an applied econometrics expert might disagree). But, you need to be very clear why you think the variables you control for may also affect whether there is mining or not. E.g. the geology associated with mining may also be associated with flat lands and fertile soils which is good for agriculture and may incentivise forest clearance for agriculture. Therefore, geology within mining municipalities is a rival explanation for forest loss. This is proxied by GVA in agriculture, or land-use change. Or industrial minings is more likely to occur in municipalities with, or close to, industrial centres which may also lead to higher GDP growth, independent of mining.

This should only require small additions to the main text to explicitly state that the confounding variables affect mining allocation too, and why. You could also add explanations of why these variables directly influence mining allocation in the Supplementary Tables 2 and 3.

Specific points:

Lines 103-106: Get rid of (control) in Line 104. Amend the following sentence to read "These estimates are based on matched samples of mining (treated) and non-mining (control) municipalities...."

Line 120: Municipalities with industrial mining showed an average direct boost ...

Line 136 – 141: My suggestion would be to delete these results or summarise in words because its quite difficult to read and readers can see that information in the Figure anyway. But up to you.

Line 154: Delete the statistic in brackets because, if I understand correctly, it is for the pre-matching data. Including it here gives it prominence and suggests it is your main result when it is not. Add 'pre-matching' to the end of that sentence to make it really clear what this sentence refers to.

Line 186: Or mining doesn't actually increase deforestation at the municipality scale...

Line 219: replace 'their contribution' with 'garimpos contributions'.

Line 253: 'Long-term strategic planning by local authorities is needed...' This is helpful to attribute responsibility. Mining companies have a responsibility to pay their taxes and abide by the laws. Governments have the responsibility to set fair laws (i.e. mandating mining companies to pay X% of profits to local authorities). Local authorities have the responsibility to ensure this money is wisely invested.

Line 334: You say that X_t is a matrix of country characteristics. Yet in Line 352 you say WX_t represents spatially-lagged regional characteristics. It seems like X_t must refer to municipality-level characteristics, otherwise they can't be spatially-lagged if they were country-level (as all municipalities would have the same value). Please correct.

Figures:

Figure 1: This is not a map of mining land cover. It is a map of Brazilian municipalities symbolised according to mined area within. Please revise the Figure legend to reflect this. For example; you could say Fig. 1: Map of mined area per municipality in Brazil.

(Remarks on code availability)

A point-by-point response to issues raised by the referees on the manuscript *Transient economic benefit and persistent forest loss: regional impacts of mining in Brazil*.

We extend our sincere gratitude to the reviewers for their constructive feedback, which has significantly enhanced the quality of our manuscript. We have diligently addressed each of the concerns raised, and implemented necessary revisions, including clearer descriptions of the methods and interpretations of results, as well as a revised title to emphasise the regional perspective of the paper.

Alongside a clean version of the revised manuscript, we enclose a version where all modifications are visibly marked. Additions to the original text are highlighted in blue, while deletions are denoted in red. Additionally, at the suggestion of the reviewers, we have added line numbers to both versions of the manuscript.

Our point-by-point response to the reviewers follows. Reviewer comments are summarised in bold and followed by direct quotations and summaries of the reviewers' concerns. The line numbers and references in our response refer to the clean version of the revised manuscript.

Answers to Reviewer #1

Comment R1.1 Use of mining intensity instead of binary indicators. *The paper uses dummies for modeling the presence or absence of mining activity. If there is data, one could also use actual mining intensity to see if that explains the effects of mining on GDP and forest loss.*

Answer to R1.1 Thank you for this comment. Initially, we considered hectares of land covered by mining as a proxy for mining intensity. However, the reason why we refrained from this as our main indicator was that it would have complicated the interpretation of our results while not substantially changing the main conclusions of the study. To stress that we have used hectares of land covered by mining as an alternative measure and to point the interested reader to the results of these robustness checks, we have adjusted the manuscript as follows:

Line 98-102: "We used the presence of industrial and garimpo mining within a mu-

municipality as the mining indicator to facilitate the interpretation of results. In the supplementary materials, we show that mining intensity in terms of mining area within each municipality does not change our main conclusions (Fig. S5 and S6)."

Line 403-405: "Results were robust with only minor exceptions against an alternative specification using mining area in ha as an indicator for the actual mining intensity (see Fig.S5 and S6)."

Comment R1.2 National-level impacts of mining on GDP. *The model is concerned with regional GDP growth. However, it is obvious that mining activities rarely benefit the local regions. Is it possible for the authors to also model the effect of mining at a national level to get a more comprehensive impact of if its positive effects?*

Answer to R1.2 Thank you for making this point. We acknowledge the significant impact mining can have on national economies and the importance of understanding the underlying mechanisms in detail. However, our study is specifically tailored to investigate the localised effects of mining in regional contexts and their immediate surroundings. A regional focus is necessary to understand how economic activities like mining benefit the local economies that constitute the concrete living environment of the population. Additionally, it helps to comprehend the spatial distribution of the associated gains and losses.

Employing a spatial econometric approach, our models are specifically designed to explore the regional impacts and spillover effects from mining municipalities to neighbouring areas. We recognise that analysing national economic impacts would necessitate different modelling approaches, likely involving time-series analyses using non-spatial data across one or multiple countries, and thus departing from the spatial panel information present in our dataset.

To emphasise the regional focus of the study, we have adjusted the title and made corresponding modifications throughout the manuscript, including the reference to a meta study on the national effects of extractive industries (Havranek et al. 2016):

Line 56-61: "By doing so, it offers a focused analysis that contrasts with broader

assessments of macroeconomic mining impacts on national indicators (28, 32). A regional focus is warranted to understand the extent to which economic activities, such as mining, benefit the local economies that constitute the concrete living environment of the population. Moreover, it is warranted as a means to understand the spatial distribution of the gains and losses from such activities.”

Line 78-82: “The Brazilian economy is widely influenced by mineral extraction. Throughout the study period, the economic growth of all Brazilian municipalities showed a strong correlation with global metals and minerals prices. However, our focus diverged from this overarching relationship between mining and the country’s economic performance as we explore regional disparities observed between mining and non-mining municipalities.”

Comment R1.3 Testing for the short-term and long-term effects of deforestation on GDP. *The effect of mining on deforestation is clear. However, does deforestation also provide a short-term positive economic benefit and perhaps a long-term negative benefit? Is this worthwhile further investigating?*

Answer to R1.3 Thank you for this comment. We agree that investigating the nexus between deforestation, mining and economic growth can prove valuable, especially with regard to exploring endogenous relationships between the three. However, while this complex issue is highly interesting, we believe that it does not fit within the scope of our study, because forest loss, in our context, is assumed to not directly affect the economy, but rather the economic activity behind forest loss, i.e. mining, may affect GDP. While there is space for lots of theoretical and methodological development, we refer the reviewer to the emerging strand of literature on the economics of deforestation, such as Silva et al. (2023), Ayad et al. (2024) and Sun et al. (2023).

Answers to Reviewer #2

The reviewer’s central critique was that the data would not support our claims made in the text. We appreciate this feedback as it highlights areas that need further clarification. We have tried to explain why we believe that our methods applied are valid and

robust. Drawing on relevant literature, we have justified our confidence in these methods, which we believe have yielded findings consistent with the conclusions drawn in the article. We have modified the manuscript accordingly to enhance clarity and coherence. In the following, we address each of the reviewer's main points of criticism individually.

Comment R2.1 National GDP trend and disparities observed between mining and non-mining municipalities. *Figure 2a shows very few differences between the GDP trends in mining and non-mining municipalities – the two track each other nearly perfectly, implying mining has very little impact on local GDP (but it could well be driving these patterns at a wider [e.g. national] scale – i.e. responsible for the trends in mining and non-mining states due to income from mining being spent at country-level).*

Answer to R2.1 Thank you for raising this point, highlighting a lack of clarity of the purpose of the descriptive figures 2a and 3a in the initial manuscript. In fact, we intended to open the results sections on GDP and forest loss, respectively, with a descriptive initial overview of differences in medians for GDP growth and forest cover loss, respectively across two distinct groups of municipalities: those with mining activity and those without. We understood that these charts would primarily showcase observed trends without factoring in underlying drivers, underscoring the necessity for a more nuanced analysis.

While we concur with the reviewer that local economic trends are typically influenced by broader national economic conditions, and that mining's impact just as well extends to the national scale, we respectfully disagree on the insignificance of the deviations observed between mining and non-mining municipalities. The highlighted differences of up to 2-4 percentage points in GDP for certain years constitute noteworthy variation from the country's yearly GDP growth rates, which have fluctuated between -4.4% and 7.5% since the 1990s (World Bank 2024).

To clarify the need for a more rigorous modelling framework capable of disentangling national from local effects and controlling for confounding factors, we have taken two key steps: (a) separating the visualisations Fig. 2a and 3a into individual figures, now designated as Fig. 2 and Fig. 4, respectively, to distinguish more clearly descriptive observations from estimation results and (b) enhancing the descriptions and clarifying the implications of the presented data in the manuscript as follows:

Line 78-93: “The Brazilian economy is widely influenced by mineral extraction. Throughout the study period, the economic growth of all Brazilian municipalities showed a strong correlation with global metals and minerals prices. However, our focus diverged from this overarching relationship between mining and the country’s economic performance as we explore regional disparities observed between mining and non-mining municipalities. Descriptive statistics suggest nuanced dynamics between these two categories, as mining municipalities tend to have outperformed non-mining municipalities during periods of ascending commodity prices and vice versa (Fig. 2). Differences were notable in 2004 and 2010, when increases in metals and minerals prices coincided with 4.0 percentage points (pp) and 3.1 pp higher GDP growth rates, respectively, in mining as compared to non-mining municipalities, and in 2015 and 2016, when mining municipalities fell back by 2.6 pp and 2.8 pp, respectively, suggesting greater vulnerability to the recessionary pressures within these areas. Nevertheless, these observations necessitated further scrutiny within a controlled modelling framework, isolating the effects of mining at the municipality level by accounting for national macroeconomic trends and potential overarching and regional confounding factors.”

Line 143-149: “Fig. 4 shows that Brazilian mining municipalities experienced higher forest loss rates than non-mining municipalities (36.7 % difference in medians between 2003 and 2020). Again, the figure merely suggests a positive link between mining and increased forest loss rates, however, without considering any hidden, potentially correlated deforestation drivers. We therefore followed the statistical approach introduced above, ensuring control over both observable and latent factors that could confound the relationship between mining and forest loss.”

Comment R2.2 Interpretation of impact estimates. Turning the attention to the presented impact estimates, Reviewer #2 argued that findings would not present distinct mining-induced effects in mining municipalities and their surroundings, as compared to all other (non-mined, non-neighbour) municipalities. Reviewer #2 stated: *Figure 2b is also concerning. Focussing on the 95% estimates shows very strong overlap both before and after 2010 between mined municipalities and their neighbours - in essence they are showing*

near identical trends. The authors claim this trend is due to mining but, since non-mined, non-neighbour municipalities are not shown on this graph, there is no evidence that other non-mined states show different trends. Indeed, Figure 2a suggests the opposite (i.e. that the non-mined states show the same trend). If the trends are the same in the mined and non-mined municipalities then it is highly unlikely to be driven by the mining! Similar statements can be made about Figure 2c and about Figures 3a, 3b and 3c. However, as the differences in trend between mined and non-mined municipalities seem greater (Figure 3a), then there is a potential impact of mining here.

Answer to R2.2 Thank you for encouraging us to clarify the applied econometric framework and the interpretation of presented results. It is correct that non-mined, non-neighbour municipalities were not shown in Fig. 2b-c and Fig. 3b-e (Fig. 3a-b and Fig. 5a-d, respectively, in the revised manuscript). However, by construction of the estimation the figures present impacts within mining municipalities (darker colours) and impacts in municipalities with mining present in their neighbouring territories (spillover effects – lighter colours) *in terms of differences* between them and non-mined, non-neighbour municipalities. These differences already account for the overarching trends observed for all Brazilian municipalities and only show effects *in addition to* national-level cycles. We have modified the manuscript and the labels of Fig. 3a-b and Fig. 5a-c accordingly to enhance clarity in interpreting the charts that present the direct and spillover impact estimates of mining.

Line 94-108: “Fig. 3 depicts mining impact estimates, i.e. the estimated differences in GDP growth in pp between mining and non-mining regions after having accounted for several observed drivers of economic growth, as well as overarching yet unobserved economic factors, such as national policy schemes and world market commodity prices that may have influenced municipalities regardless of their exposure to mining (see Methods for details). [...] Each panel shows median impact estimates surrounded by 95 % certainty intervals for the respective years or multi-year periods (pooled impacts for pre-2010 and post-2010 time frames) in two colours. The darker colour represents direct impacts within mining municipalities and the lighter colour indicates the magnitude of spillover effects of mining across municipality borders. All impacts indicate estimated average differences as compared to non-mining, non-neighbouring municipalities, keeping all other municipality characteristics in the model constant.”

Line 152-154: “Note that the figure mirrors the presentation of Fig. 3, displaying direct and spillover impact estimates within 95 % uncertainty intervals.”

To clarify that our modelling framework takes into account a range of additional factors that potentially determine Brazilian municipalities’ GDP and forest loss, we have incorporated explanations in the Methods section, detailing that the broader trends observed in the time series, such as the apparent synchronicity between mining and non-mining municipalities (noted in comment R2.1), are accounted for in our models through the inclusion of annual fixed effects:

Line 285-289: “Incorporating time-specific fixed effects, for example, accounts for factors influencing the dependent variables in specific years of the sample. The models can thereby consider trends affecting municipalities regardless of their exposure to mining, such as national (e.g., macroeconomic conditions, environmental and economic policies) and global (e.g., commodity price fluctuations) factors.”

Comment R2.3 Robust impact evaluation. Reviewer #2 argues that robust impact evaluation required a proper counterfactual. Reviewer #2 states: *I am not an expert in the methods the authors have applied, and so I encourage the editors to seek out a methodological expert in impact evaluation. However, whilst I am not convinced the mined vs non-mined municipal comparison shows any meaningful difference (see above), I think the comparison is oversimplistic. Robust impact evaluation requires a proper counterfactual and it is unlikely that the non-mined municipalities provide this. The methods would be vastly improved by using statistical matching to show that some of the non-mined municipalities provide a good counterfactual or, indeed, if they do not, then perhaps constructing a synthetic control. A BACI is the gold standard in impact evaluation and should be performed here as it would allow for other potential differences to be accounted for (e.g. wider changes in GDP driving by national policy that impact the mined municipalities and the counterfactual identically). I suspect that once these have been accounted for, there will be no difference between the GDP in mined municipalities and their neighbours, nor with non-mined municipalities either! If so, this would lead to a very different conclusion to that drawn by the authors.*

Answer to R2.3 Thank you for this comment. We acknowledge that there is a vast set of methodological approaches for economic and environmental impact assessment and we see the advantages of the methods raised. However, given the available data and our aim of considering spillover effects, we determined that a spatial econometric framework was most appropriate for this study. While we refrain from making strong causal claims, understanding that other study designs might be more suitable for that purpose, we believe our chosen method is valid and well-suited for demonstrating the local economic and environmental impacts of mining. As mentioned, our method accounts for differences in municipality characteristics without relying on exact counterfactuals or data pruning through matching procedures. For the reasoning behind the full set of control variables, please refer to the Data section. This approach is commonly used in econometric panel studies, as referenced in our manuscript.

To clarify our reasoning for selecting this statistical approach, we have made revisions to the Introduction and the Methods section of the manuscript. Moreover, we hope that the changes addressed in our answer to R2.2 enhance the clarity and understanding of our results. Thank you again for your valuable feedback.

Line 65-70: “Besides its advantage of explicitly accounting for and evaluating the spatial spillovers that are central to this work, our spatial econometric approach offered a robust statistical framework that considered various national and local drivers of regional GDP and forest loss. This approach contrasts with experimental approaches or before-after control-impact assessments, which need careful monitoring of both impact and control groups over time.”

Line 263-272: “To assess the economic and environmental impacts across the entire Brazilian mining sector, we adopted a well-established econometric approach, building on the extensive literature that examines the effects of natural resource extraction on socio-economic indicators (32). This empirical methodology, also used to analyse the determinants of economic growth (48, 49) and the causal pathways of deforestation (50, 51), is well-suited for investigating macro-level dynamics at the regional scale. It allowed us to isolate the effects of mining by controlling for various observable and unobservable confounding factors, i.e. factors affecting both

the presence of mining and the outcomes of interest (economic growth and forest loss), thereby also accounting for dynamics that may influence both mining and non-mining municipalities.”

Line 273-276: “In light of the potential effects of mining activities on neighbouring areas, we aimed to account for the spatial spread of impacts across geographical locations. Combining well with the spatial nature of the mining data at hand, this study employed a spatial econometric approach [...]”

Comment R2.4 Lack of line numbers in the manuscript. *Due to a lack of line numbers in the manuscript, it is very difficult to give feedback on more minor errors within the manuscript.*

Answer to R2.4 Thank you. We have added line numbers to the revised versions, and we would greatly appreciate any feedback on errors you may find.

References

Ayad, Hicham et al. (2024): Assessing deforestation in the Brazilian forests: An econometric inquiry into the load capacity curve for deforestation. In: *Forest Policy and Economics* 159, 103135. DOI: [10.1016/j.forpol.2023.103135](https://doi.org/10.1016/j.forpol.2023.103135).

Havranek, Tomas/Horvath, Roman/Zeynalov, Ayaz (2016): Natural Resources and Economic Growth: A Meta-Analysis. In: *World Development* 88, 134–151. DOI: [10.1016/j.worlddev.2016.07.016](https://doi.org/10.1016/j.worlddev.2016.07.016).

Silva, Jonathan Gonçalves da/Almeida, Roselaine Bonfim de/Carvalho, Leandro Vinicios (2023): An economic analysis of a zero-deforestation policy in the Brazilian Amazon. In: *Ecological Economics* 203, 107613. DOI: [10.1016/j.ecolecon.2022.107613](https://doi.org/10.1016/j.ecolecon.2022.107613).

Sun, Luxi et al. (2023): Deforestation embodied in global trade: Integrating environmental extended input-output method and complex network analysis. In: *Journal of Environmental Management* 325, 116479. DOI: doi.org/10.1016/j.jenvman.2022.116479.

World Bank (2024): World Development Indicators. <https://data.worldbank.org/indicator/NY.GDP.MKTP.KD.ZG?end=2022&locations=BR&start=1990>.

A point-by-point response to issues raised in the second round of the referees on the manuscript *Uncertain economic gains and forest loss: Evidence from industrial and garimpo mining in Brazilian municipalities*.

We appreciate the thoughtful and constructive feedback provided by the four reviewers, which has helped improve the clarity and robustness of the manuscript. In response, careful revisions have been made to the analysis and presentation. Specifically, a matching approach has been implemented, and results have been re-estimated using matched data to mitigate bias and confounding. We have also refined the wording around causality and results reporting to enhance clarity and ensure alignment with the evidence, including a revised title. To further clarify the study design and prevent misunderstandings, we have added an overview figure illustrating the workflow (see Fig. S10 in the supplementary materials).

To facilitate the review process, both a clean version of the revised manuscript and a tracked-changes version are provided, where additions are highlighted in blue and deletions in red. The line numbers and references in our response refer to the clean version of the revised manuscript.

A detailed point-by-point response follows. Since multiple reviewers raised concerns about the study design – particularly the distinction between identifying causal effects and demonstrating correlations – these comments are addressed collectively first, followed by individual responses to the remaining points.

We sincerely appreciate the time and effort you have taken to review this work.

Identification of impacts vs. demonstrating correlations

- Several reviewers raised concerns regarding the study design, particularly the distinction between identifying causal effects and demonstrating correlations. Reviewer 2 argued that the language used to describe causality was too strong given the empirical framework and provided specific examples from the manuscript.

– **Comment R2.1.** *Thank you for these revisions - they have improved the manuscript and make it much clearer. I still have some concerns relating to the method used - namely that before-after control-impact assessments would be a more appropriate method to use. As the authors acknowledge in their response letter, our study designed are more appropriate for*

making strong causal claims. The authors indicate that they therefore refrain from these claims. However, I do not think that is the case and, as such, without amendments, the manuscript is misleading - i.e. makes claims that are not supported by the results due to the methods used. [...] The text throughout the manuscript needs editing to ensure language such as that highlighted above are not used, and the manuscript very clearly refers to correlations and not 'impact'.

- Reviewer 3 acknowledged that the paper is “well written and that the applied problem is well positioned in the literature,” but also expressed concerns that the study design does not sufficiently support causal inference. However, R3 also noted a lack of expertise in the specific methods applied.

– **Comment R3.1.** *While in the response to reviewers they respond to the query about the methods they use to draw causal conclusions by claiming not to make strong causal claims in the paper, the paper is framed as exploring and demonstrating causal relationships. I don't see how they could downplay this-if they are simply demonstrating an association, then the paper is not interesting. However, I am not convinced that the study design is sufficient to base causal claims on and I do not feel that they sufficiently acknowledge the assumptions underlying their identification strategy (or provide support for these assumptions). However, these comments come with a warning that I do not completely follow their approach (I have not used the sort of spatial model they use myself).*

– **Comment R3.2.** *Causal claims in this context (which they do make, despite their comment in the response to reviewers) depend on the assumption that once they have controlled for observed confounders, that allocation of municipalities to the mining or no mining group is as if random. It is very difficult to believe this assumption is true especially given that they do not appear to do any matching to ensure the set of municipalities included in the analysis is at all similar. One concern I had was that they are presumably including in the analysis both municipalities with almost no forest cover and ones with lots of forest cover. A low loss of forest in terms of hectares per municipality area might be because the municipality has little forest, or because little has been lost. If I have missed that their model accounts for this I am sorry (it is possible but I am not clear where).*

Answer to R2.1, R3.1, R3.2: Thank you for raising these important issues. We have carefully considered these concerns and recognise the challenges involved. The nature of our data and the study's objective – to derive general insights beyond municipal borders and to cover the entire country, allowing some indication of the direction of effects to be generalisable – does

not easily lend itself to an experimental study design (which would be the best to discover causalities). However, while macroeconometric regional studies do not always explicitly frame their analyses in such terms, before-after comparisons are somewhat incorporated through the panel setting. The binary indicator for mining status shifts from 0 (not present) to 1 (present) over time, and in some cases, back to 0 when mining activity ceases. Combined with a range of control variables, we believe that this approach already extends beyond simple correlations. That said, we fully understand the concerns regarding the assumption that mining assignment should be as random as possible across the sample. We decided to address the concerns in two ways: (1) We applied a matching procedure to make our study more robust, as it helps reduce potential bias from factors that could influence whether a municipality receives the treatment (mining), and overall, reduces model dependence. The results after matching suggest that our findings extend beyond simple correlations. However, we acknowledge that this is not a controlled experimental study, and matching is only one step towards a more robust causal analysis. (2) Recognising the remaining uncertainty, we have carefully adjusted the wording to ensure the results are presented more as associations rather than causal effects.

Matching approach

We employed coarsened exact matching (CEM) to address potential biases in our analysis. Specifically, we performed matching separately for the industrial mining and garimpo mining samples, and subsequently used the resulting pruned data as inputs for the respective econometric models.

The matching process did not alter the main conclusions of our paper. However, it did result in smaller absolute forest loss estimates, likely due to the exclusion of exceptionally large municipalities or otherwise incomparable units during the matching process. To facilitate comparison between the results from the previous submission and those obtained with the updated matching approach, we have included figures at the end of this document.

We have updated the manuscript accordingly, incorporating a mention of the matching procedure in the Introduction (lines 75-78) and providing a more detailed description along with relevant references in the Methods section (lines 528-556). Additionally, we have included Table S4 and Fig. S11 presenting the matching results in the supplementary materials.

In response to R3.2's concern regarding initial forest cover, we clarify that this variable is accounted for in our analysis. Initial forest cover was included as a control in the previous models, and we have now added a comprehensive table in the supplementary materials listing all control variables and explaining their inclusion (see our replies in the next section). Addition-

ally, initial forest cover is considered in the matching procedure, ensuring better comparability between municipalities before addressing other confounders in the econometric models. This step reduces bias from differences in initial forest conditions between the treatment and control groups by excluding municipalities with large disparities. Any remaining imbalance is addressed by incorporating variables such as initial forest cover as controls.

Language

We have revised the wording in several areas throughout the document, including changes to the title and subtitles in the Results section, in response to the reviewers' feedback. In particular, following one reviewer's suggestion, we have opted to use the term "effects" instead of "impacts" in both the figures and text, recognising that this may help avoid stronger associations with causality, while still conveying the intended relationships. Below are some examples of the changes made:

Line 1-3: "Uncertain economic gains and forest loss: Evidence from industrial and garimpo mining in Brazilian municipalities"

Line 86- : "GDP trends in mining regions"

Line 152: "Mining and forest loss patterns"

Line 11-13: "For less regulated "garimpo" mining concessions, we identify substantial associations with elevated deforestation rates, highlighting the environmental risks of insufficient oversight."

Line 54-55: "This study investigates the contribution of mining to deforestation and regional economic growth in Brazil at the municipality level."

Line 78-84: "Our findings indicate substantial associations between garimpo mining and increased deforestation, while industrial mining shows inconclusive connection to forest cover dynamics. Regarding regional economic outcomes, the potential for positive effects appear more pronounced for industrial mining. However, neither garimpo nor industrial

mining prompts reliable or lasting increase in local GDP, with industrial mining even linked to negative effects in some years.”

Line 103-104: “Fig. 3 presents estimates of the relative effect of mining, expressed as the differences in GDP growth (in pp) between mining and non-mining (control) regions.”

Potential confounders and considered control variables

Several reviewers raised concerns regarding the potential confounders in our analysis and the clarity of the control variables considered.

- Reviewer 3’s feedback:
 - **Comment R3.3.** *I would have really liked to see a table showing the confounding variables their design takes account of (and the datasets used to capture them). A DAG would also be helpful.*
 - **Comment R3.4.** *In any analysis such as this, the question is always whether there are likely to be hidden confounders which are large enough to disrupt the results. I can think of plenty potential confounders. A solid hidden confounder analysis would be valuable (though I am not sure how it can be implemented with this model, as I say I am not familiar with it).*
 - **Comment R3.5.** *Finally, I find the claim that the design shows spill over rather unconvincing. Could not the same hidden confounders drive patterns seen?*
- Reviewer 4’s feedback:
 - **Comment R4.1.** *I agree with some of the concerns of Reviewer 2. Without a path diagram/theory of change outlining your assumptions I am not certain that you have controlled for all possible observable confounders of the causal relationship between mining and GDP growth/forest cover change. I agree with the other reviewer that there could be outstanding differences between municipalities with and without mining which could confound your estimates. For instance, perhaps mining municipalities have specific geological characteristics which affect soil productivity and agricultural development, and consequently GDP and forest cover. Having looked at Supplementary Table 1 I think that many of your variables probably do capture or proxy for many possible confounders but as authors with much greater knowledge of the study area than me, I think its your responsibility to make this crystal clear. So, I would like to see an explanation (probably in SI due to word limits and briefly*

outlined in the main text) of the potential confounders, and how your data capture or proxy for these. If you think there is a reasonable chance that there remain unobserved confounders, you could include a statement that your results reflect association between the presence of mining and outcomes but not causality.

Answer to R3.3, R3.4, R3.5, R4.1: We sincerely appreciate these comments and, in particular, the encouragement from Reviewer 4 to clarify how our model effectively accounts for key confounding factors. To enhance clarity, we have revised the data section to provide a more structured explanation of the included variables and the rationale behind their selection. It now features dedicated subsections on dependent variables, mining indicators, covariates and potential confounders, and statistical matching. Additionally, please note that the application of matching as explained above further strengthens the analysis by improving comparability between municipalities with and without mining, addressing potential differences more effectively.

Regarding the suggestion to include a DAG or additional tables, we opted to provide tables detailing the rationale behind our covariate selection (Tables S2 and S3), complementing Table S1, which lists variable definitions and data sources. A DAG was considered but ultimately not included, as the complexity of potential interactions made it challenging to represent in a linear, clear and interpretable manner. Instead, we reference the recent work by Lo et al. (2024), which follows a similar logic in variable selection.

We acknowledge that no observational study can entirely eliminate unobserved confounding, but we believe our approach is robust due to the panel design, large sample size, and inclusion of time-fixed effects, which control for substantial unobserved heterogeneity. To explicitly reflect this, we have added the following statement:

Line 521-527: “While we have carefully accounted for a comprehensive set of covariates, it is important to acknowledge that residual confounding may still persist. Unobserved or complex factors, which are difficult to capture in the model, could influence the relationships between mining, GDP, and forest cover. Nevertheless, the inclusion of time-fixed effects and the longitudinal panel design that tracks outcomes over time, helps mitigating the potential for such unobserved heterogeneity (61). Moreover, the large sample size and temporal coverage of the data add confidence in the reliability of our estimates.”

Regarding the identification of spillover effects, our spatial econometric model accounts for the confounding influence of neighbouring observations more effectively than a non-spatial approach. A key strength of this model is its ability to explicitly capture spatial dependen-

cies by incorporating spatially lagged design matrices (WX and $W\tilde{X}$) and spatially lagged dependent variables (Wy and $W\tilde{y}$) into the regression equations. This ensures that both the characteristics within a given municipality and those of its neighbouring units are considered in the estimation. Additionally, we now elaborate on how mining interacts with other economic activities, reinforcing broader land-use transformations and resource exploitation dynamics, which in turn influence forest cover:

Line 233-238: “The pronounced spillover effects further suggest that garimpo mining is embedded within broader land-use transformations that contribute to deforestation. Rather than occurring in isolation, garimpos may accelerate existing processes such as logging, agriculture, and the expansion of infrastructure into previously undisturbed areas, reinforcing cycles of land degradation and resource exploitation.”

Answers to Reviewer #1

Comment R1.1 Reviewer 1 suggests that the paper should explore whether deforestation, driven by mining, has a positive or negative economic impact, as the focus on both mining’s economic effects and deforestation feels disconnected. *The authors have responded to my comments; however, I still believe that the impact of mining on deforestation has not been convincingly addressed. The paper finds a substantial impact of mining on deforestation. Therefore, the logical next step would be to ask whether the economic impact of this deforestation has been positive or negative, which the authors have not addressed in the revised version. If deforestation is not at all a channel through which the economy is impacted, why dedicate a major section of the paper to it? If the paper aimed solely to estimate the impact of mining on deforestation, it would make sense to leave it as it is. However, since the goal is to highlight the impact of mining on the local economy and deforestation, it would make sense to test whether areas with deforestation have seen any adverse impacts on their agricultural GDP, natural resources, etc. At the moment, the paper appears somewhat disjointed, with one aspect exploring the impact of mining on the economy and the other exploring the impact on deforestation. Spending some effort on linking the two would benefit the paper and better inform the readers, in my opinion.*

Answer to R1.1 Thank you for raising this important point. We completely agree that investigating the relationship between deforestation and GDP is a valuable avenue for future research. However, this paper focuses on two distinct aspects: the effects of mining on GDP and its effects on forest cover. The results and discussion are structured to address these two dimensions

separately, with the discussion connecting them to highlight relevant policy implications. We have chosen to follow a similar approach to a recent study on nickel mining in Indonesia (Lo et al. 2024), which also investigates these aspects separately. Exploring how deforestation affects GDP would require different models, specifically addressing the endogeneities within the triangle of mining, forest loss, and economic growth. While this is not feasible within the current paper, it remains an important direction for future research.

Answers to Reviewer #3

Comment R3.6 *This paper is on very important topic (the impacts of mining on deforestation and local economy), I was therefore very interested to read this paper. The paper is well written and the applied problem is well positioned in the literature. The data and code are available which is great to see. This is clearly a resubmission and I read the thoughtful response to reviewers the authors had prepared. This was all good. They made appropriate changes.*

Answer to R3.6 Thank you!

Comment R3.7 *Figure 1 is really helpful. It presents really useful data and helps us understand its structure. Tiny point: you could label the regions more clearly, I struggled with them.*

Answer to R3.7 Thank you for your feedback. We have adjusted the figure to enhance clarity by using a brighter red border to highlight the Legal Amazon and Minas Gerais regions, making them more distinguishable while maintaining the figure's overall balance.

Comment R3.8 *Y axis isnt labelled in Fig 4*

Answer to R3.8 Axis label added!

Answers to Reviewer #4

Comment R4.2 *Thank you for this submission. It is a really interesting study and I enjoyed reading it. I have three main areas of concern which currently prevent me from recommending that this article be accepted. However, they mostly relate to the communication of results and conclusions and therefore should be a relatively easy fix.*

Answer to R4.2 Thank you!

Comment R4.3 Is there a reason why you haven't used unit-fixed effects to control for time-invariant heterogeneity?

Answer to R4.3 Thank you for this comment. We opted not to use unit-fixed effects because the spatial Durbin model (SDM) already accounts for spatial dependence, which can partially capture unobserved heterogeneity across units. Additionally, introducing unit-fixed effects in this setting may lead to incidental parameter bias, particularly given our sample size and model complexity. We have added reference 61 (Elhorst 2010) in line 526 to provide interested readers with more details on fixed effects in spatial panel models.

Comment R4.4 Are you sure that you have controlled for factors that may uniquely affect GDP growth rate in the control municipalities but not the mining ones?

Answer to R4.4 Thank you for this comment. Yes, we have controlled for key determinants of GDP growth and forest loss as established in the literature, ensuring that our specification captures the most relevant environmental, economic, and sociodemographic factors. While unobserved heterogeneity is always a concern, the inclusion of these standard controls minimizes the risk of omitted variable bias. For further details, please refer to the section "Potential confounders and considered control variables" in this document.

Comment R4.5 Language. *I note that in your previous rebuttal letter you say that you have been careful about making causal claims and I think this is good, given that there may still be unobserved confounders. However, I disagree with some of the reporting of your results. I am not convinced that your results support the language you use to report your main findings. In the more detailed points below I have highlighted instances where I think your language is too strong. For example: You show that municipalities with industrial mining are associated with lower economic growth than those without for 3 years (2010, 2011 and 2012). The number of years with a negative effect is exceeded by those with a positive effect, and the maximum positive effect is higher than the maximum negative one. Yet, in the introduction you state that the associated boost in GDP is transient, however, I would argue that the associated drop in GDP growth is transient as you show that the positive effect of mining recovers from 2014. No-one ever likes to tone down the strength of their main message but I think that with the suggested amendments your main message still stands that the economic contributions from mining are*

not guaranteed given they are dependent on global commodity prices and therefore vulnerable to shocks. However, you need to make sure your reporting aligns with your results.

Answer to R4.5 Thank you for your thoughtful feedback and for encouraging a more balanced framing of our findings. We have carefully revised several sections of the manuscript to better align with the message that the economic contributions of mining are inconsistent and unreliable. This is reflected in the revised title, as well as in several adjustments throughout the text. For instance, we have made the following changes:

Line 13-15: “The economic benefits of mining are limited. Particularly for industrial mining, they are tied to fluctuations in global mineral prices.”

Line 78-84: “Our findings indicate substantial associations between garimpo mining and increased deforestation, while industrial mining shows inconclusive connection to forest cover dynamics. Regarding regional economic outcomes, the potential for positive effects appear more pronounced for industrial mining. However, neither garimpo nor industrial mining prompts reliable or lasting increase in local GDP, with industrial mining even linked to negative effects in some years.”

Line 188-193: “Our results suggest that fluctuations in global commodity prices not only contributed to economic stagnation and crisis in Brazil but also reshaped the observed relationship between mining and regional economic output, at times even reversing it. Before 2010, a favourable global economic environment, marked by high commodity prices and strong material demand was associated with higher economic growth in mining municipalities and their neighbouring regions compared to similar non-mining areas.”

Line 203-210: “However, the expansion of the extractive sector can undermine other, potentially more sustainable economic activities, such as small-scale agriculture or manufacturing, while increasing dependence on mining (26). Our results demonstrate that this dependence makes regional economies vulnerable to fluctuations in global commodity prices. When prices decline, the same backward linkages that once underpinned growth become pathways for economic contraction, affecting not only mining municipalities but also neighbouring regions. Over time, this volatility can undermine broader development efforts and

reinforce economic instability.”

Comment R4.6 R4 raises concerns about indirect effects of deforestation found: *I think a confounding factor could be affecting the estimates of spillover of deforestation from garimpo mining, because I am struggling to believe the magnitude of effects. I can think of processes through which the spillover is working in the opposite direction a deforestation frontier develops in neighbouring municipalities, more people arrive and spread into adjacent municipalities where they then discover gold, leading to the expansion of deforestation for garimpo mining. In this case the potential causal relationship goes the other way; increasing deforestation in neighbours is a proxy for frontier development and in-migration which then causes the discovery of gold in the mined municipality. In the other direction, indirect deforestation associated with mining is usually concentrated relatively close to the mining area (e.g. within 5km: <https://www.nature.com/articles/s41893-024-01421-8>). Unless all the mining areas are very close to municipality borders, I'm not convinced that this cause enough spillover deforestation to cause such a large effect. I suppose the frontier effect could be at play and people attracted by mining could cross municipality borders and clear land for farming/pasture, but in that case I find it strange that mining wouldn't also expand. Particularly given that the gold exploited by garimpo miners are usually secondary deposits along river beds which can extend long distances, crossing municipality boundaries. If you believe this pattern of spillovers detected in your results does reflect the reality in Brazil, please explain this in text and the potential mechanisms behind this effect. If you think it might be spurious, please also discuss this.*

Answer to R4.6 Thank you for this comment. We agree that the spread of mining, especially garimpos, and their interactions with other economic activities create complex dynamics. While we control for various land uses and leverage a longitudinal sample, fully isolating effects remains challenging. The literature on indirect deforestation effects of mining is still limited – while the cited study provides one perspective of effects at rather short distances, others, such as Sonter et al. (2017), have found spillovers extending up to 70 km. To better reflect these complexities, we have revised the manuscript as follows:

Line 233-238: “The pronounced spillover effects further suggest that garimpo mining is embedded within broader land-use transformations that contribute to deforestation. Rather than occurring in isolation, garimpos may accelerate existing processes such as logging, agriculture, and the expansion of infrastructure into previously undisturbed areas, reinforcing cycles of land degradation and resource exploitation.”

The possibility that deforestation, in turn, creates opportunities for mining is an important consideration, introducing endogeneity that is challenging to fully eliminate. While this challenge cannot be entirely avoided, we mitigate potential confounding through measures such as matching, which has notably reduced the detected spillover effects (see figures at the end of the document). Importantly, in contrast to other studies, our spatial econometric framework explicitly captures spillovers, providing valuable insights into their presence. While the magnitude of these effects may also depend on the construction of spatial weights, our findings clearly demonstrate their existence, highlighting the need for further research into spatial dependencies and the broader spread of mining-related change.

Comment R4.7 *Are you sure that the non-mining municipalities never mined?*

Answer to R4.7 Thank you for this comment. We rely on the high-quality data provided by MapBiomas, which is based on advanced remote sensing techniques. While some margin of error is inevitable, non-mining in our study means that in the respective year, no mining activity was detected in the satellite imagery – this is the most reliable classification available. We cannot rule out the possibility that mining occurred before our sample period, but given the large panel structure of our data and controlling for a range of “initial” conditions in the municipalities, this is unlikely to affect our results significantly.

Comment R4.8 *Lines 10–15: See comment above. I disagree with this statement. Given that the positive effect of industrial mining on GDP growth recovers in 2014 and 2015, I think the negative effect is transient. I recommend you talk instead about the effect of industrial mining on GDP growth being inconsistent and unreliable, as it is vulnerable to global mineral prices. Based on the evidence presented here I don't think you can say that Brazil's extractive industries have failed to deliver lasting economic advantages. If looked at differences over the whole time period 2005–2015 you might find that mining did have an overall positive effect, or at least not a negative one. So please delete this sentence. However, you can ask questions about the sustainability and reliability of mining-based economic growth. I also think you should separate the sentence on the effects of garimpo mining on deforestation from the sentence about the effects of industrial mining on GDP growth. Otherwise it sounds like you are talking about the same type of mining when you are not.*

Answer to R4.8 Thank you for this comment. We have revised the abstract accordingly, refining the wording and separating the statements to improve clarity, such that:

Line 11-17: “For less regulated “garimpo” mining concessions, we identify substantial associations with elevated deforestation rates, highlighting the environmental risks of insufficient oversight. The economic benefits of mining are limited. Particularly for industrial mining, they are tied to fluctuations in global mineral prices. These findings challenge the perception that mining inherently drives sustained regional economic development.”

Comment R4.9 *Line 11: I also wonder if you should say less regulated garimpo mining concessions for those who are familiar with small-scale mining (although the editors might disagree!) Or less regulated small-scale mining concessions.*

Answer to R4.9 We have incorporated this suggestion as well – please see our revisions above.

Comment R4.10 *Line 3: I think you need to mention governance when talking about the resource curse because this is the main reason why in some countries, resource wealth has not translated to GDP growth and poverty reduction. Lack of investment in health, education and manufacturing is a symptom of that.*

Answer to R4.10 Thank you. We have incorporated this into the manuscript as follows:

Line 47-52: “However, this perspective is countered by research on the “resource curse thesis”, which highlights the potential negative economic consequences of resource wealth. Studies suggest that in contexts of weak governance and institutional quality, an abundance of natural resources may hinder the development of key sectors such as manufacturing, education, and health – sectors which are vital for sustained growth (27-31).”

Comment R4.11 *Line 55: Use the term artisanal and small-scale mining here as this is a key term for this sort of mining.*

Answer to R4.11 We have revised the manuscript accordingly:

Line 58-60: “Moreover, the present study differentiates between industrial and artisanal and small-scale mining, known as “garimpos” (Fig. 1d), both provided for in Brazilian legislation (Supplementary Text A).”

Comment R4.12 *Line 102: replace impact with effect. Also, Fig. 3 caption says that you are plotting the posterior means, yet in the text it says median? Then below in lines 107 you say average too. I think you are plotting the average difference, so please change median to average.*

Answer to R4.12 Thank you for pointing this out. We have adjusted the text to use 'average' instead of 'median'. Moreover we have replaced 'impacts' with 'effects' throughout the manuscript, including in the figures. For example, the revised paragraph introducing Fig. 3 now reads as follows:

Line 111-117: "[...] Each panel shows mean effect estimates surrounded by 95 % certainty intervals (CI) for the respective years or multi-year periods (pooled effects for pre-2010 and post-2010 time frames) in two colours. The darker colour represents direct effect within mining municipalities and the lighter colour indicates the magnitude of spillover effects of mining across municipality borders. All effects indicate estimated average differences as compared to non-mining, non-neighbouring municipalities, keeping all other municipality characteristics in the model constant."

Comment R4.13 *Line 101: Ambivalent is not the correct term to describe this pattern (I don't think a positive effect in 7 years compared to 2 with 2 with a negative effect qualifies as ambivalent). Please use varied or variable over time. Don't say across Brazil's 5262 municipalities because it sounds like the effect varied over space, rather than time.*

Answer to R4.13 We have revised the text accordingly, as follows:

Line 118-120: "The findings in Fig. 3a suggest that the local economic effects of Brazilian industrial mining varied over time, with notable differences between the periods before and after 2010."

Comment R4.14 *Line 124: Change uncertainty to estimates*

Answer to R4.14 We have revised the manuscript accordingly:

Line 133-134: “The years since 2010 showed more variable results, with estimates ranging from positive to negative values in several years.”

Comment R4.15 *Lines 152 154: Unnecessary and confusing, please delete*

Answer to R4.15 Thank you for pointing this out. We have removed the statement as suggested.

Comment R4.16 I guess those estimates of 894 ha per year of direct deforestation from garimpo is the estimated total across all mined municipalities? I am struggling to believe the magnitude of spillover effects and the mechanism behind it here. Usually indirect deforestation associated with artisanal and small-scale mining is relatively close to the mined area (i.e. within 5km, see (<https://www.nature.com/articles/s41893-024-01421-8>)). See main point #3 above. Please add an explanation if you believe this is a real effect or potentially spurious.

Answer to R4.16 Yes, the interpretation of 894 ha is correct. Please note that this figure has been updated following the application of matching before running the regressions. Regarding the indirect effects, please refer to our response to comment 4.4.

Comment R4.17 *Lines 167 169. This is very interesting!*

Answer to R4.17 Please note that in the updated version, after applying matching, the results are less robust against alternative specifications. This issue remains intriguing and certainly warrants further investigation, especially as large mining companies, such as those in Carajás, often argue that they are protecting surrounding forests by ensuring that their concessions – which are much larger than the actual mining areas – are designated as protected from other uses. We have revised the paragraph, such that:

Line 179-186: “In certain years, as well as in the pooled post-2010 estimates, municipalities with industrial mining activity appeared to have a protective effect on neighbouring forest cover, both in absolute and relative terms (Figs. 5a,c). However, this observed effect is not supported by an alternative model specification that replaces binary mining indicators with ha of mining area (Fig. S5). The alternative model instead suggests that forest loss spillovers from industrial mining are comparable to those associated with garimpo mining. Within

municipalities directly hosting industrial mines, the evidence remains inconclusive, leaving it uncertain whether industrial mining mitigates or exacerbates forest loss.”

Comment R4.18 *Line 174: please dont use the term causing because you have shown association, not causality.*

Answer to R4.18 Thank you for your comment. As a general approach, we have aimed to avoid causality-related terminology in our revisions. The revised version of this section is as follows:

Line 188-190: “Our results suggest that fluctuations in global commodity prices not only contributed to economic stagnation and crisis in Brazil but also reshaped the observed relationship between mining and regional economic output, at times even reversing it.”

Comment R4.19 *Lines 194 196: Please amend this statement. Some garimpo profits will feed back into the local economy through local spending by miners, traders and those further down the supply chain. I totally agree that the fact that the gold is traded informally probably contributes to the evidence of no effect on GDP, but I dont think you say that no profits are spent locally.*

Answer to R4.19 Thank you for pointing this out. We agree that garimpos may play an important socioeconomic role and have revised the statement accordingly, such that:

Line 211-220: “Compared to industrial mining, garimpos showed weaker associations with economic growth, and we found no periods where effect estimates turned negative. One possible explanation lies in their frequently informal or illegal nature, which allows profits to evade official record-keeping or be transferred out of the region. At the same time, garimpo mining provides a livelihood for an estimated 200,000 people in the Brazilian Amazon (39), many of whom have limited economic opportunities. While GDP-based analyses may understate the local economic significance of garimpos, their broader socioeconomic role remains insufficiently understood. Future research should explore alternative, more granular measures of regional well-being to provide a more comprehensive assessment of their contribution to regional economies.”

Comment R4.20 *Lines 213 215: If you are going to talk about banning garimpo gold mining then you*

need to say that this will have serious impacts on the livelihoods of miners, many (but not all) of whom I guess are very poor. Perhaps a better suggestion would be to focus on the need for better regulation and support to improve environmental practices (there is lots of literature on this from this from which you can draw examples). Also because there is heaps of evidence from around the world that bans just dont work, and often end up making things worse by pushing miners into more remote areas and increasing conflict.

Answer to R4.20 Thank you for this comment. As reflected in the revisions made in 4.19, we acknowledge the socioeconomic importance of garimpos, particularly for the very poor. We have removed the controversial claim on halting garimpo gold mining and instead adopted a more constructive framing, incorporating additional sources. We appreciate the suggestion and agree that this would be an interesting topic to explore further. The revised manuscript now states:

Line 238-242: “These findings highlight the urgent need for effective and enforceable policies to close regulatory loopholes, uphold environmental standards, and support improvements in the environmental performance of garimpos (42, 43). Formalising garimpos could play a key role in this effort, enabling better traceability of mined materials and ensuring compliance with environmental safeguards, such as mercury-free gold processing (44).”

Comment R4.21 *Lines 223: stronger negative impacts than positive impacts is not supported by your data, please delete.*

Answer to R4.21 Thank you for pointing this out. This misunderstanding has been resolved, and the revised version now reads as follows:

Line 248-250: “While mining may benefit municipalities and their neighbours during boom phases, local mining economies can also experience bust phases associated with economic setbacks.”

Comment R4.22 *Lines 225 227: long-term downturns (as discussed in the resource curse literature. Long-turn downturns sounds like it is related to economic crises/commodity price drops and I dont think the resource curse literature particularly focusses on this (but I could be wrong). I think its best to just delete this and say creating resilience to downturns is essential to guarantee the economic stability of*

mining operations.

Answer to R4.22 Thank you for your comment. We have followed your suggestion, and the revised version now reads as follows:

Line 252-253: “Creating resilience to downturns is essential to guarantee the economic stability of mining operations.”

Comment R4.23 *Lines 230–231: I think that this statement contradicts your findings. Apart from the 4 years where mining has a negative or zero effect, for all the other years you show that industrial mining had a positive effect on GDP growth. And you state that local economies constitute the concrete living environment of the local population – so do benefit citizens. Please rewrite this sentence to be more specific. Perhaps you could say that the positive socio-economic effects could be greater with a more equal distribution of economic benefits, and that this could help to buffer the effect on local economies during shocks.*

Answer to R4.23 Thank you for your comment and suggestion. We have revised the manuscript accordingly, and the updated version now reads as follows:

Line 257-261: “Socioeconomic benefits could be strengthened through a fairer distribution of mining revenues – rather than the largest shares accruing to mining companies – and improved efficiency and transparency in the allocation of financial transfers to subnational and local governments (45). Moreover, long-term planning is needed to address the socio-economic prospects of communities in mined-out areas.”

Comment R4.24 *Line 257: while likely failing economic promises I think this is too strong. Please tone down. You could say something like while making local economies dependent on mining. This increasing vulnerability to external shocks (e.g. in mineral prices), and raises questions about the sustainability of economic development.*

Answer to R4.24 Thank you for your comment. We have toned down the statement, and the revised version now reads as follows:

Line 281-286: “To conclude, our findings support the concerns raised earlier that the contin-

ued expansion of the extractive sector can increase deforestation and related pressures on Brazilian forests, particularly the Amazon, while deepening local economic dependence on mining (15, 16, 49, 50). This dependence heightens vulnerability to external shocks, such as fluctuations in mineral prices, and raises questions about the long-term stability and sustainability of regional development.”

Comment R4.25 *Lines 315–316: This sentence could do with more explanation.*

Answer to R4.25 Thank you for pointing this out. We have added more information, and the revised version now reads as follows:

Line 341–344: “We selected 2010 as the separation point based on our yearly coefficient results, which revealed significant pattern shifts, including a marked change in industrial mining GDP estimates and the conclusion of a period with particularly high absolute forest loss estimates for garimpos.”

Comment R4.26 *Lines 470–473: I’m not sure that the average GDP growth rate would capture all other anthropogenic drivers of land use change. What about smallholder agriculture?*

Answer to R4.26 Thank you for your comment. We agree that GDP growth alone cannot capture all anthropogenic drivers, but we believe it serves as a useful control for economic activity. When combined with land use and land use change variables, as summarised in the new Table S3, it provides a solid framework. These controls, based on land use satellite observations, also address smallholder agriculture quite effectively.

References

- Elhorst, J. Paul (2010): Spatial Panel Data Models. In: Handbook of applied spatial analysis. Ed. by Manfred M Fischer/ Arthur Getis. Berlin, Heidelberg: Springer, 377–407.
- Lo, Michaela G.Y. et al. (2024): Nickel mining reduced forest cover in Indonesia but had mixed outcomes for well-being. In: One Earth 7(11), 2019–2033. DOI: [10.1016/j.oneear.2024.10.010](https://doi.org/10.1016/j.oneear.2024.10.010).
- Sonter, Laura J. et al. (2017): Mining drives extensive deforestation in the Brazilian Amazon. In: Nat. Commun. 8(1), 1013. DOI: [10.1038/s41467-017-00557-w](https://doi.org/10.1038/s41467-017-00557-w).

Comparison of new (using matching) and old results

Response to Reviewers – Third Round of Review

Manuscript Title: *Forest loss and uncertain economic gains: Evidence from industrial and garimpo mining in Brazilian municipalities*

We would like to thank the reviewers and editors for their continued engagement with our manuscript. We appreciate the time and care taken to provide detailed feedback across the review rounds. The comments have been very helpful in further clarifying and improving the paper.

In this latest revision, we have carefully addressed all points raised. To support the review process, we are submitting both a clean version of the manuscript, formatted according to the journal's requirements, and a tracked-changes version. The latter highlights all revisions (additions in blue, deletions in red) and includes line numbers, which we refer to throughout this response. Please note that Figures 1 to 5 have also been updated in accordance with the journal's artwork guidelines (<https://www.nature.com/documents/aj-artworkguidelines.pdf>).

Below, we respond point by point to the reviewers' comments, explaining the changes made and referencing specific parts of the manuscript where relevant.

Answers to Reviewer #4

Comment R4.1 Thank for your kind and thoughtful response to my last comments and the substantial changes you have made to the manuscript. It is much improved now.

Answer to R4.1 Many thanks for your kind words and for your thoughtful and fair feedback throughout the review process. We are glad to hear that the changes have strengthened the manuscript.

Comment R4.2 Thank you for changing the title too – it is much better aligned with your results now. Can I just check though – do you intend to say that the forest loss from mining is uncertain? If not, consider switching the title to read: 'Forest loss and uncertain economic gains:...'

Answer to R4.2 Thanks for pointing this out – and for the helpful suggestion. Our intention was to reflect the uncertainty surrounding the economic gains from mining, not to suggest that forest loss is uncertain. On the contrary, the link between mining and forest loss is often clearly observable – particularly in regions like the Brazilian Amazon, where satellite imagery reveals

forest disturbance with striking clarity (and the question is rather for which type of mining it applies, and what's the size of the effect). To avoid any confusion, we have changed the title as you suggested: "Forest loss and uncertain economic gains: Evidence from industrial and garimpo mining in Brazilian municipalities". We appreciate your close reading.

Comment R4.3 I still think how you present the confounding factors could be improved. In the section Covariates and potential confounding factors you haven't explicitly highlighted which variables you think are correlated with the presence of mining and causally effect GDP and forest loss (i.e. the confounding factors). This is made more confusing by the fact that in Supplementary Tables 2 and 3 it is only discussed how the variables influence outcomes, not treatment at all. In causal inference studies aiming to ascertain the effect of an intervention you only need to control for, via matching and regressions, hypothesized confounding factors. If you think precipitation only influences forest loss and not the presence of mining, you don't need to control for it because you are saying it is okay to assume that mining is randomly allocated along the precipitation gradient, therefore precipitation will not affect average differences in outcomes between treated and control units.

However, in complex land systems sometimes while a factor might not directly affect allocation to treatment it could be correlated with something else that does, meaning allocation to treatment is not as if random across this gradient. For example, I can see in Figure S11 that there were differences in precipitation between mining and non-mining mining municipalities (although still relatively small). For that reason, I think it's probably okay to include non-directly confounding factors in the regressions, just in case (although an applied econometrics expert might disagree). But, you need to be very clear why you think the variables you control for may also affect whether there is mining or not. E.g. the geology associated with mining may also be associated with flat lands and fertile soils which is good for agriculture and may incentivise forest clearance for agriculture. Therefore, geology within mining municipalities is a rival explanation for forest loss. This is proxied by GVA in agriculture, or land-use change. Or industrial minings is more likely to occur in municipalities with, or close to, industrial centres which may also lead to higher GDP growth, independent of mining.

This should only require small additions to the main text to explicitly state that the confounding variables affect mining allocation too, and why. You could also add explanations of why these variables directly influence mining allocation in the Supplementary Tables 2 and 3.

Answer to R4.3 Thank you for the detailed feedback on confounding factors. We agree that the earlier version focused more on how covariates influence the outcome variables, rather than

addressing their potential to affect both the treatment and the outcomes. There are several factors such as biophysical, economic, and demographic variables that can influence whether mining is likely to occur or expand in a given area. As you pointed out, competition for land between different uses, as well as factors like accessibility, play a key role. In light of this, we have clarified the potential confounding factors in the revised version of the “Covariates and potential confounding factors” section, as outlined below. In addition, we have added clarifications how these variables may influence mining allocation in Supplementary Tables 2 and 3.

Lines 452-554: “We accounted for a broad set of covariates in the design matrices X and \tilde{X} to better isolate the effect of mining on GDP and forest cover by controlling for confounding factors that could simultaneously influence these outcomes and the presence of mining.”

Lines 463-567: “Such patterns act as useful proxies for underlying factors, including fertile soils, residential development, or conservation, which may confound our analysis by simultaneously influencing mine development (e.g., fertile soils may attract land uses that compete with mining) and the outcome variables of our interest. ”

Lines 501-505: “The role of the labour market in shaping mining activity is similarly ambiguous and likely context-dependent. Higher levels of human capital may attract industrial mining by reducing training costs and increasing productivity, while simultaneously discouraging informal or artisanal mining, as more educated populations are likely to pursue alternative livelihoods.”

Lines 528-531: “These variables help account for heterogeneities in economic growth and forest cover dynamics, as well as underlying processes such as land competition and local material demand, which may either facilitate or constrain the development of mining activities.”

Lines 542-545: “These biophysical characteristics capture key spatial constraints that may influence GDP growth and forest cover, as well as factors relevant to mining presence or expansion, such as accessibility, land development potential, and susceptibility to wildfires or erosion.”

As you correctly noted, some of the variables we considered may not be direct confounders, but rather determinants of the outcome variables where it is not clear how and if at all they influence treatment. To address this, we have added a statement explaining why we still believe it is valuable to control for them. Including these variables helps to increase the precision of our effect estimates:

Lines 484-487: “While some of these variables are not obvious confounders – i.e. they may influence the dependent variables without affecting mining expansion, or they may only exert an indirect influence via other channels – their inclusion is expected to improve the precision of effect estimates by accounting for additional variation in the outcomes.”

We hope that these revisions have clarified the presentation of confounders and, once again, thank you for your detailed and helpful feedback, which has contributed significantly to improving the paper.

Comment R4.4 Lines 103-106: Get rid of (control) in Line 104. Amend the following sentence to read “These estimates are based on matched samples of mining (treated) and non-mining (control) municipalities...”

Answer to R4.4 Thank you for the close reading. We have revised the manuscript accordingly.

Comment R4.5 Line 120: Municipalities with industrial mining showed an average direct boost...

Answer to R4.5 Thank you for pointing this out. We have revised the sentence to include “municipalities with” as suggested.

Comment R4.6 Line 136-141: My suggestion would be to delete these results or summarise in words because its quite difficult to read and readers can see that information in the Figure anyway. But up to you.

Answer to R4.6 Thank you for this helpful suggestion. We agree that the paragraph was quite dense and have revised it to improve readability by summarising key trends in clearer terms. However, we decided to retain some of the numerical detail to provide readers with concrete

reference points that complement the visual information in the Figure.

Comment R4.7 Line 154: Delete the statistic in brackets because, if I understand correctly, it is for the pre-matching data. Including it here gives it prominence and suggests it is your main result when it is not. Add ‘pre-matching’ to the end of that sentence to make it really clear what this sentence refers to.

Answer to R4.7 Thank you for pointing this out. We agree, and have therefore removed the statistic in brackets and rephrased the sentence to clarify that it refers to comparison of the raw data:

Lines 159-161: “Fig. 4 shows that Brazilian mining municipalities experienced higher forest loss rates than non-mining municipalities in the pre-matching data.”

Comment R4.8 Line 186: Or mining doesn’t actually increase deforestation at the municipality scale...

Answer to R4.8 Thank you for pointing this out. We agree that this is an important nuance and have revised the sentence such that:

Lines 191-194: “Within municipalities directly hosting industrial mines, the evidence remains inconclusive, suggesting that industrial mining may not have a consistent effect on forest loss at the municipal scale.”

Comment R4.9 Line 219: replace ‘their contribution’ with ‘garimpos contributions’.

Answer to R4.9 Thank you for the suggestion. We have revised the sentence accordingly and included the possessive apostrophe to reflect the plural form:

Lines 226-228: “Future research should explore alternative, more granular measures of regional well-being to provide a more comprehensive assessment of garimpos’ contributions to regional economies.”

Comment R4.10 Line 253: ‘Long-term strategic planning by local authorities is needed’ This is helpful to attribute responsibility. Mining companies have a responsibility to pay their taxes and abide by the laws. Governments have the responsibility to set fair laws (i.e. mandating mining companies to pay X% of profits to local authorities). Local authorities have the responsibility to ensure this money is wisely invested.

Answer to R4.10 Thank you for this comment. We have revised the sentence accordingly to clearly attribute responsibility.

Comment R4.11 Line 334: You say that X_t is a matrix of country characteristics. Yet in Line 352 you say WX_t represents spatially-lagged regional characteristics. It seems like X_t must refer to municipality-level characteristics, otherwise they can’t be spatially-lagged if they were country-level (as all municipalities would have the same value). Please correct.

Answer to R4.11 Absolutely – thank you for spotting this mistake. We have corrected the text accordingly to:

Lines 343-344: “ X_t is an $n \times k$ matrix of k exogenous municipality characteristics in the initial period.”

Comment R4.12 Figure 1: This is not a map of mining land cover. It is a map of Brazilian municipalities symbolised according to mined area within. Please revise the Figure legend to reflect this. For example; you could say Fig. 1: Map of mined area per municipality in Brazil.

Answer to R4.12 Thank you for pointing this out. We have revised the caption to more accurately reflect the figure content. The updated title (in bold) summarises what panels (a)-(d) have in common, while the descriptions of the panels were clarified accordingly. We retained the phrase *area classified as mining (18)* to acknowledge that the data are based on a remote sensing classification, as well as to indicate the source. The revised caption now reads:

*Lines 808-814: “Fig. 1: **Mining area by municipality and aggregated totals in Brazil, 2005-2020.** Map of total area classified as mining (18) per municipality in 2020 (in ha), with the Legal Amazon and Minas Gerais outlined in red (a); average annual change in mining area within municipalities between 2005 and 2020 (in %) (b); mining area by region (2005-2020,*

in 1,000 ha) **(c)**; and mining area by mining regulation type (2005-2020, in 1,000 ha) **(d)**.”